EMBO
Molecular Medicine

# Am80-GCSF synergizes myeloid expansion and differentiation to generate functional neutrophils that reduce neutropenia-associated infection and mortality

Lin Li[1,2,†], Xiaotian Qi[3,†], Weili Sun[4,5], Hisham Abdel-Azim[4,5], Siyue Lou[1], Hong Zhu[1], Nemani V Prasadarao[5,6], Alice Zhou[1], Hiroyuki Shimada[1,5], Koichi Shudo[7], Yong-Mi Kim[4,5], Sajad Khazal[4], Qiaojun He[2], David Warburton[3,5] & Lingtao Wu[1,5,*]

## Abstract

Neutrophils generated by granulocyte colony-stimulating factor (GCSF) are functionally immature and, consequently, cannot effectively reduce infection and infection-related mortality in cancer chemotherapy-induced neutropenia (CCIN). Am80, a retinoic acid (RA) agonist that enhances granulocytic differentiation by selectively activating transcription factor RA receptor alpha (RARα), alternatively promotes RA-target gene expression. We found that in normal and malignant primary human hematopoietic specimens, Am80-GCSF combination coordinated proliferation with differentiation to develop complement receptor-3 (CR3)-dependent neutrophil innate immunity, through altering transcription of RA-target genes RARβ2, C/EBPε, CD66, CD11b, and CD18. This led to generation of functional neutrophils capable of fighting infection, whereas neutralizing neutrophil innate immunity with anti-CD18 antibody abolished neutrophil bactericidal activities induced by Am80-GCSF. Further, Am80-GCSF synergy was evaluated using six different dose-schedule-infection mouse CCIN models. The data demonstrated that during "emergency" granulopoiesis in CCIN mice undergoing transient systemic intravenous bacterial infection, Am80 effect on differentiating granulocytic precursors synergized with GCSF-dependent myeloid expansion, resulting in large amounts of functional neutrophils that reduced infection. Importantly, extensive survival tests covering a full cycle of mouse CCIN with perpetual systemic intravenous bacterial infection proved that without causing myeloid overexpansion, Am80-GCSF generated sufficient numbers of functional neutrophils that significantly reduced infection-related mortality in CCIN mice. These findings reveal a differential mechanism for generating functional neutrophils to reduce CCIN-associated infection and mortality, providing a rationale for future therapeutic approaches.

Keywords functional neutrophils; innate immunity; myeloid expansion; neutrophil bactericidal activity; neutrophil differentiation
Subject Categories Cancer; Immunology; Pharmacology & Drug Discovery

## Introduction

CCIN is a condition in which the number of neutrophils in patients' bloodstream is decreased, leading to increased susceptibility to microbial infections. Thus, rapidly generating large amounts of functional neutrophils through "emergency" granulopoiesis (Panopoulos & Watowich, 2008) is needed to defend against infection in CCIN. GCSF, a lineage-specific hematopoietic growth factor for promoting neutrophil production from hematopoietic stem cells (HSC) (Panopoulos & Watowich, 2008), has been a main drug used for CCIN treatment for over 2 decades. The increased expansion of granulocytic precursors in response to GCSF results in an elevated proportion of early neutrophil forms *in vivo*, for example, myeloblast or promyelocyte (Lord *et al*, 1989, 1991), whereas neutrophils induced by GCSF are inadequately differentiated with impaired microbicidal functions (Leavey *et al*, 1998; Donini *et al*, 2007; Dick

1   Department of Pathology, Children's Hospital Los Angeles Saban Research Institute, Los Angeles, CA, USA
2   Institute of Pharmacology and Toxicology, Zhejiang University, Hangzhou, Zhejiang, China
3   Developmental Biology and Regenerative Medicine Program, Children's Hospital Los Angeles Saban Research Institute, Los Angeles, CA, USA
4   Pediatric Hematology-Oncology, Blood and Marrow Transplantation, Children's Hospital Los Angeles Saban Research Institute, Los Angeles, CA, USA
5   University of Southern California Keck School of Medicine, Los Angeles, CA, USA
6   Division of Infectious Diseases, Children's Hospital Los Angeles Saban Research Institute, Los Angeles, CA, USA
7   Japan Pharmaceutical Information Center, Shibuya-ku, Tokyo, Japan
   *Corresponding author. Tel: +1 323 361 6318; Fax: +1 323 361 3669; E-mails: lwu@chla.usc.edu; lwu@therapeuticapp.org
   †These authors contributed equally to this work

*et al*, 2008; Ding *et al*, 2013). Therefore, although GCSF profoundly increases the number of immature neutrophils to reduce the duration of neutropenia, CCIN remains challenging with substantial morbidity and mortality (Hartmann *et al*, 1997; Heath *et al*, 2003; Sung *et al*, 2007; Bohlius *et al*, 2008; Gurion *et al*, 2012; Mhaskar *et al*, 2014), as well as healthcare cost (Liou *et al*, 2007; Michels *et al*, 2012). Moreover, there is a potential association between clinical use of GCSF and development of myeloid malignancy (Smith *et al*, 2003; Rosenberg *et al*, 2006; Hershman *et al*, 2007; Socie *et al*, 2007; Beekman & Touw, 2010). As such, an alternative mode of treatment is needed for CCIN, which can induce sufficient numbers of functional neutrophils while preventing the risk of myeloid malignancy.

RA has potent pan action for binding to and transactivating two classes of nuclear receptor transcription factors, retinoic acid receptors (RARα, β, γ) and retinoid X receptors (RXRα, β, γ) (Chambon, 1996; Umemiya *et al*, 1997; Kagechika, 2002). RA induces granulocytic differentiation by mediating gene transcription in various normal and tumor tissues (Collins, 2008; Gudas, 2012), for example, transactivating RARγ to enhance HSC self-renewal while promoting differentiation of committed myeloid progenitors by activating RARα (Purton *et al*, 2006). Am80 (tamibarotene) (Kagechika, 2002) is designed to avoid the side effects of RA pan action in clinical use, for example, skin irritation resulting from RA binding of RARγ (Standeven *et al*, 1997). Although Am80 is selective for RARα and RARβ, which function as the master regulators of granulocytic differentiation and tumor suppression, respectively (Alvarez *et al*, 2007; Collins, 2008), it preferentially activates RARα (Umemiya *et al*, 1997; Kagechika, 2002; Jimi *et al*, 2007). Since 2005, Am80 has been approved for treatment of acute promyelocytic leukemia (APL) in Japan (Miwako & Kagechika, 2007). Recent studies show that Am80 can induce modest production of neutrophils displaying significantly higher bactericidal activity *in vitro*, compared to those induced by GCSF (Ding *et al*, 2013). This raises question: Could Am80-GCSF combination serve as a novel synergist, acting through differential mechanisms by synchronizing Am80's competence for promoting neutrophil differentiation with GCSF's profound capability for myeloid expansion, to induce sufficient numbers of functional neutrophils fighting infection?

Using different human granulopoietic systems and mouse CCIN models, our study reveals a novel Am80-GCSF synergy that can induce large amounts of functional neutrophils to reduce CCIN-associated infection and mortality while preventing myeloid overexpansion. This is likely through altering transcription of RA-target genes that promote differentiation of GCSF-expanded granulocytic precursors into functional neutrophils with well-developed CR3 immunity.

# Results

## Am80-GCSF alters transcription of RA-target genes that promote induction of large amounts of neutrophils with well-developed innate immunity in normal primary human hematopoietic specimens

We first asked whether Am80-GCSF combination could induce more functional neutrophils than did Am80 alone. Normal primary human hematopoietic CD34$^+$ precursors were treated with human plasma concentration doses of Am80 and/or GCSF (Appendix Table S1) for up to 6 days. Then, neutrophils generated from CD34$^+$ cells were evaluated with bacterial killing as well as production of reactive oxygen species (ROS), an essential neutrophil function against infection (Nauseef, 2007; Uriarte *et al*, 2011). The results (Fig 1A) showed that compared to Am80, Am80-GCSF induced significantly larger amounts of neutrophils with better morphologic differentiation, increased bacterial killing, as well as enhanced ROS production stimulated with neutrophil activator fMLP or PMA (Barnes *et al*, 2012). Since Am80 selectively activates RARα (Umemiya *et al*, 1997; Kagechika, 2002; Jimi *et al*, 2007) to alter transcription of cell cycle inhibitor *p21*$^{Cip/Kip}$ versus several other granulocytic differentiation regulators in CD34$^+$ precursors and NB4 cells (Appendix Fig S1), we investigated how Am80-GCSF modulated expression of 12 different RA-target genes (Appendix Table S2) that play key roles in control of granulocytic differentiation. CD34$^+$ cells were treated with Am80-GCSF for different time periods and then followed by qRT–PCR analyses. We found that 6 of 12 genes analyzed were dynamically modulated by Am80-GCSF at different differentiation induction stages (Fig 1B, day 1 versus day 2 versus day 6), including the tumor suppressor *RARβ₂* (Soprano *et al*, 2004; Alvarez *et al*, 2007), terminal granulocytic differentiation regulator *C/EBPε* (Park *et al*, 1999; Lekstrom-Himes, 2001), as well as neutrophil innate immunity regulators *CD66c, CD66b, CD11b* (Skubitz *et al*, 1996; Park *et al*, 1999;

---

**Figure 1.  Am80-GCSF alters transcription of RA-target genes that promote induction of large amounts of neutrophils with well-developed innate immunity in normal primary human hematopoietic specimens.**

A   Proliferation analysis of hematopoietic CD34$^+$ precursors for up to 6 days (i). Neutrophil morphologic differentiation, bacterial killing, and ROS production were assessed at day 6 (ii–iv). Controls were without treatment. White arrows indicate neutrophil nuclear segmentation.

B   qRT–PCR-assessed RA-target gene expression after culturing CD34$^+$ cells for 1, 2, and 6 days.

C   Flow cytometry analysis of CD66-CD18 co-expression after culturing CD34$^+$ cells for 6 days, using anti-human CD66-PE and CD18-FITC antibodies (i). Isotypes were used for controls (i, top panel). CD66-CD18 co-expression was quantified in (ii).

D   Fresh peripheral blood (PB) was collected from normal human donor. Bacterial killing was assessed in the presence or absence of PB neutrophils (i). Bactericidal activities of neutrophils induced by Am80-GCSF from PB mononuclear cells were evaluated by ROS production (ii) and bacterial killing assays (iii).

E   Fresh PB neutrophils collected from normal human donor were tested for ROS production in the presence or absence of specific antibodies. Controls were with or without IgG. Comb Ab, combined anti-CD18 and -CD11b antibodies.

F   Fresh PB mononuclear cells collected from normal human donors were treated for 3 days. Bacterial killing (i) and neutralization of bacterial killing in the presence of IgG or anti-CD18 antibody were assessed at day 3 (ii).

Data information: Data are shown as mean ± SD and represent at least two independent experiments with similar results. \*P < 0.05; \*\*P < 0.01; \*\*\*P < 0.001 (Student's unpaired two-tailed *t*-test). Exact *P*-values are provided in Appendix Table S4. A+G, Am80-GCSF combination; CL, chemiluminescence; RLU, relative light units; AUC, area under the curve; φ, normal IgG; Ab, antibodies.

Ding *et al*, 2013), and *CD18* (Bush *et al*, 2003; Ding *et al*, 2013). Notably, whereas both *RARβ2* and *C/EBPε* were consistently induced, they were associated with differential transcriptional induction of innate immunity regulators, that is, *CD66c* in all stages, *CD66b* in the early stage, *CD11b* in the middle stage, and *CD18* in the late stage, suggesting that Am80-GCSF mediated a course of

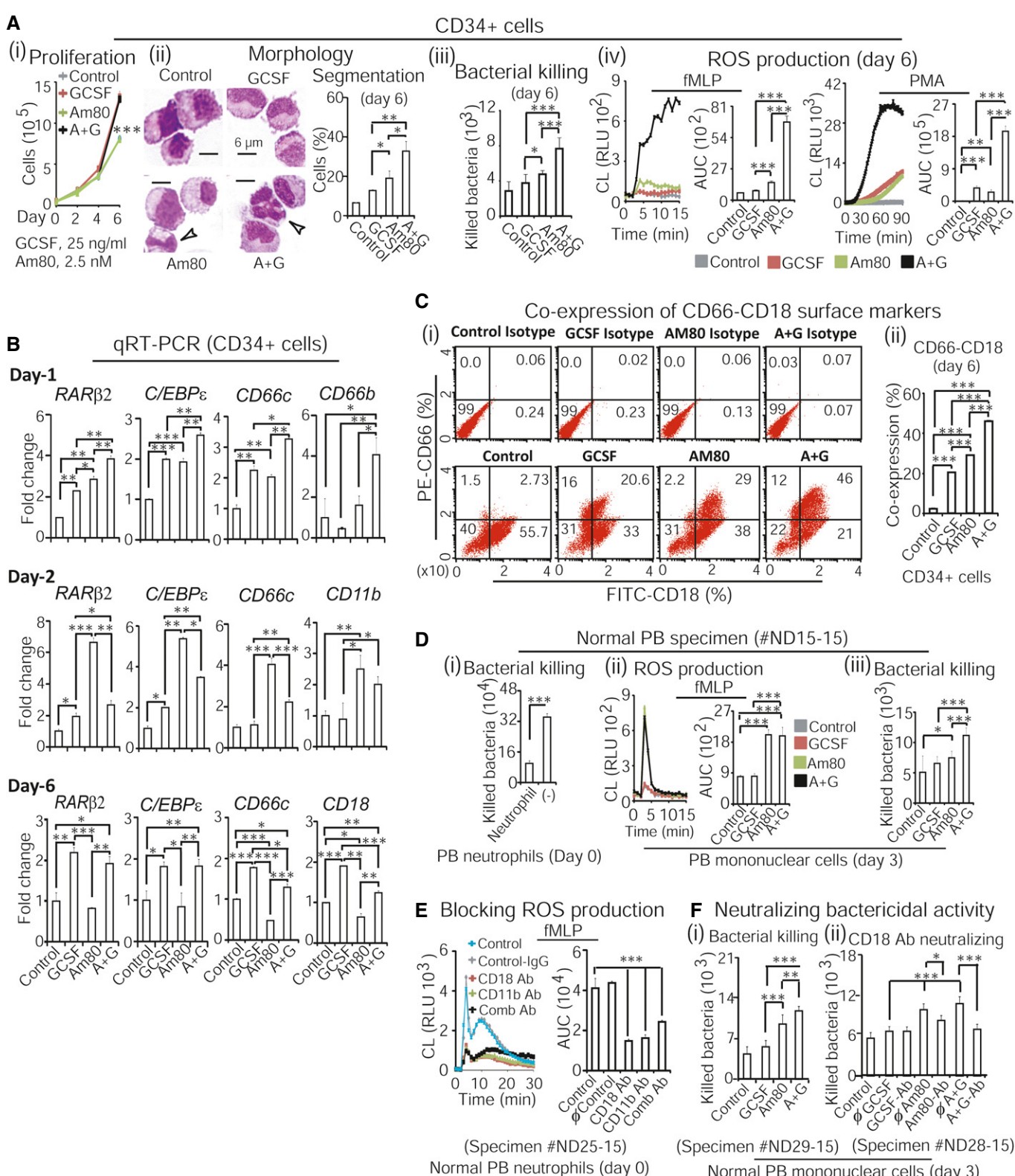

**Figure 1.**

neutrophil differentiation-associated innate immunity development. Markedly, Am80-GCSF promoted significantly higher expression of these genes than did Am80 in the early and late differentiation induction stages (Fig 1B, day 1 and day 6), whereas Am80 induced higher expressions in the middle stage (Fig 1B, day 2). Interestingly, although both GCSF and Am80-GCSF are highly statistically significant in promoting transcription of $RAR\beta_2$, $C/EBP\epsilon$, $CD66c$, and $CD18$ than did Am80 in the late differentiation induction stage, such effects on changes in transcriptional induction folds were relatively small (Fig 1B, day 6). It is known that co-expression of different CD66 subunits with CD18 surface marker are crucial in mediating CR3-dependent neutrophil innate immunity against infection (Skubitz *et al*, 1996; Ding *et al*, 2013). Thus, we assessed co-expression of CD66-CD18 surface markers induced by Am80-GCSF in the late differentiation induction stage, using flow cytometry analyses. The results showed that Am80-GCSF induced significantly higher co-expression of CD66-CD18 surface markers than did Am80 or GCSF alone at day 6 (Fig 1C). Moreover, because CD18 and CD11b are CR3 components mediating neutrophil function against microbial infection (Groves *et al*, 2008), we tested whether CR3-dependent bactericidal activities induced by Am80-GCSF in $CD34^+$ cells were also promoted in primary human peripheral blood (PB) specimens collected from normal human donors. The results showed that compared to GCSF, neutrophils induced by Am80-GCSF from normal primary human PB mononuclear cells showed significantly higher bacterial killing and ROS production, mimicking the bactericidal activities observed in normal primary human PB neutrophils (Fig 1Di versus ii, iii). Further, anti-CD18 and/or -CD11b antibodies markedly abolished ROS production in normal primary human PB neutrophils (Fig 1E). Accordingly, neutrophils differentiated from normal primary human PB mononuclear cells by Am80-GCSF killed markedly more bacteria, whereas such increased bacterial killing was significantly blocked with anti-CD18 antibody neutralization (Fig 1Fi versus ii). These data show that Am80-GCSF can generate significantly larger amounts of functional neutrophils than does Am80, whereas significantly higher expression of CD66, CD18, and CD11b induced by Am80-GCSF in these neutrophils promotes the development of neutrophil innate immunity.

## Significant induction of $RAR\beta_2$ and $C/EBP\epsilon$ by Am80-GCSF is associated with both production of functional neutrophils and growth inhibition in primary human leukemic specimens

Systematic review and meta-analysis of 5,256 patients show that providing GCSF to acute myeloid leukemia (AML) patients post-chemotherapy does not affect overall survival or infectious rate (Gurion *et al*, 2012), whereas GCSF may induce myeloid malignancy in neutropenic patients (Smith *et al*, 2003; Rosenberg *et al*, 2006; Hershman *et al*, 2007; Socie *et al*, 2007; Beekman & Touw, 2010). Thus, we investigated whether Am80-GCSF combination could generate functional neutrophils against infection while suppressing malignant growth in non-APL AML specimens. We first combined low human plasma concentration dose of Am80 (20 nM) with medium human plasma concentration dose of GCSF (25 ng/ml) for the test, as described in Appendix Table S1. The results showed that in treatment of bone marrow (BM) specimen collected from AML patient for 4 days, Am80-GCSF induced significantly higher ROS

production than did GCSF or Am80 while sustaining proliferation similar to GCSF (Fig 2Ai and ii). Further to assess dynamic changes in expression of RA-target genes in cultured AML specimen, we found that similar to those in normal cells (see Fig 1B), Am80-GCSF induced significantly higher transcription of RA-target genes compared to Am80 in both early and late differentiation induction stages (Fig 2Bi, day 2, day 6). However, in contrast to normal cells (see Fig 1B), Am80-GCSF induced significantly higher expression of RA-target genes than did GCSF in AML cells throughout the full differentiation induction period (Fig 2Bi, days 2, 3, 6), including tumor suppressor $RAR\beta_2$ (Soprano *et al*, 2004; Alvarez *et al*, 2007), terminal granulocytic differentiation regulator $C/EBP\epsilon$ (Park *et al*, 1999; Lekstrom-Himes, 2001), and CR3 component $CD11b$ (Park *et al*, 1999). Although this altered transcription pattern by Am80-GCSF was associated with significantly higher ROS production than Am80 or GCSF, Am80-GCSF sustained proliferation similar to GCSF in AML specimens with 3 days of treatment (Fig 2Bii versus iii).

Next, using a longer period of treatment, we titrated dose ranges of Am80 when combined with GCSF in generating functional neutrophils while inhibiting leukemic growth in primary AML specimens. The clinical doses of GCSF for CCIN treatment have been established worldwide over the past 2 decades, whereas the medium human plasma concentration dose of GCSF (25 ng/ml) in mediating granulocytic differentiation of different human hematopoietic precursors (Appendix Table S1B) has been well established *in vitro* (Hao *et al*, 1998; Luo *et al*, 2007; Lou *et al*, 2013). Therefore, we evaluated different Am80 doses that ranged from low to high human plasma concentrations converted from clinical use, including 20, 50, 100, and 150 nM Am80, in combination with 25 ng/ml GCSF, respectively (Appendix Table S1). First, BM specimens collected from AML patients were treated with 20 or 50 nM Am80 alone or in combination with GCSF for 9 days. The results showed that compared to GCSF, these Am80-GCSF combinations significantly inhibited proliferation after 9 days of treatment (Fig 3Ai and ii). We then performed 12 days of treatment using 20, 100, or 150 nM Am80 in combination with GCSF. We found that similar to the observation derived from 9-day treatment (see Fig 3A), the combination of 100 or 150 nM Am80 with GCSF significantly inhibited proliferation compared to GCSF alone in all specimens tested (Fig 3Bi–iv). However, the combination of 20 nM Am80 with GCSF did not inhibit leukemic growth in one specimen (Fig 3Biv). Moreover, neutrophils induced by 150 nM Am80 combined with GCSF killed significantly more bacteria than those induced by GCSF or Am80 alone (Fig 3Ci and ii). Further, we systemically evaluated the effect of 150 nM Am80, the plasma concentration dose nearly reaching the routine clinical usage of Am80 for APL (Appendix Table S1A), on inducing functional neutrophils when combined with GCSF. The results showed that in parallel to inhibiting leukemic growth after 12 days of treatment (Fig 3Di), Am80-GCSF promoted the highest neutrophil ROS production, bacterial killing, and morphologic differentiation (Fig 3Dii–iv). Notably, compared to GCSF or Am80 alone, Am80-GCSF markedly induced both $RAR\beta_2$ and $C/EBP\epsilon$ in association with $CD66c$ (Fig 3Dv). Similarly, in the NB4 leukemia cell line, Am80-GCSF also induced growth inhibition, ROS production, and altered RA-target gene expression (Appendix Fig S2). Collectively, these data show that by sustaining a consistently significant induction of both tumor suppressor $RAR\beta_2$ and terminal

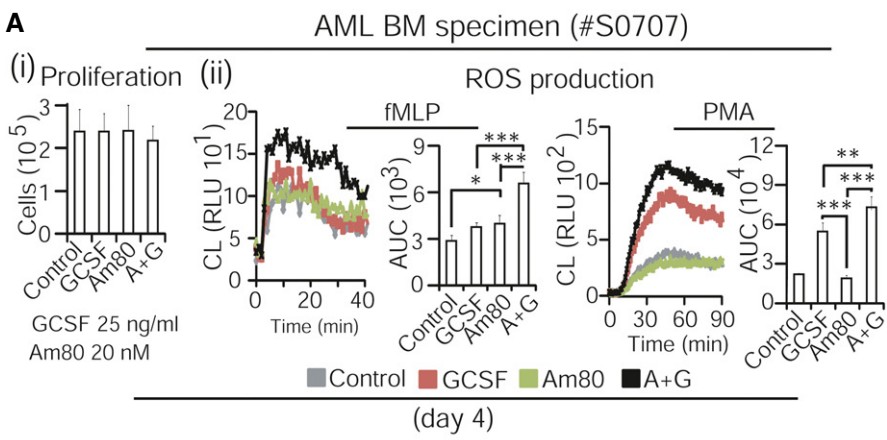

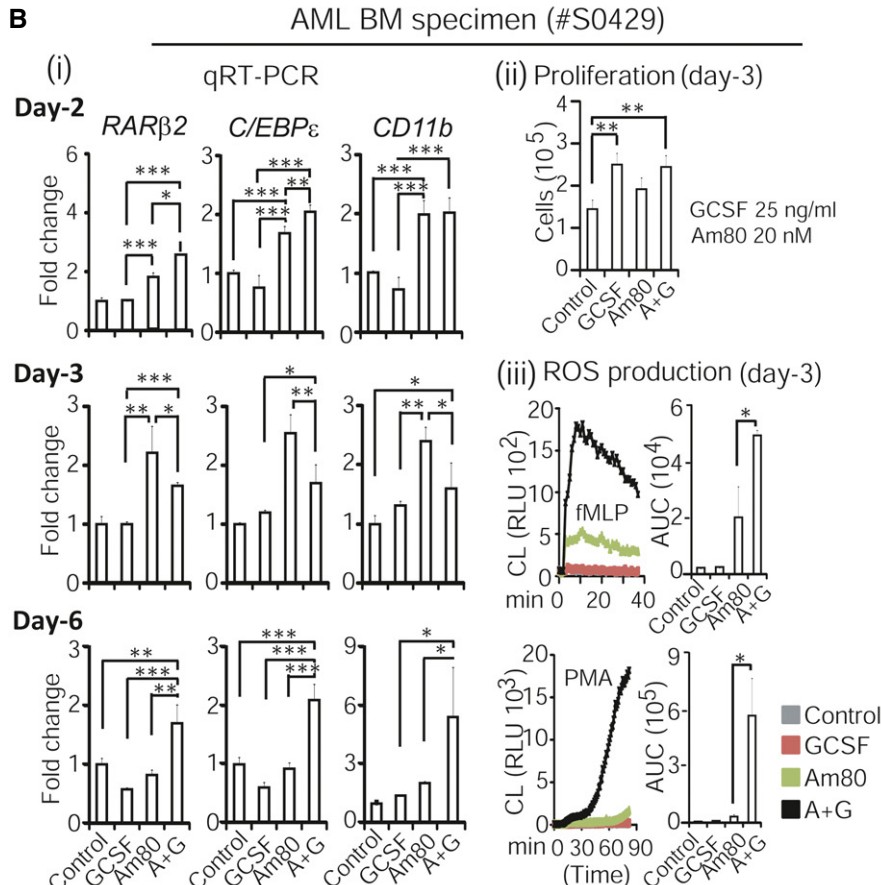

**Figure 2. Am80-GCSF induces both higher bactericidal activities and transcription of RA-target genes that regulate neutrophil differentiation in primary human leukemic specimens.**

A   Fresh bone marrow (BM) mononuclear cells were collected from acute myeloid leukemia (AML) patient. Proliferation (i) and ROS production (ii) were analyzed after cells were treated for 4 days.

B   Fresh BM mononuclear cells collected from AML patient were treated for up to 6 days. Transcription of RA-target genes was analyzed with qRT–PCR at day 2, 3, and 6 (i). Both proliferation and ROS production were assessed at day 3 in parallel (ii, iii).

Data information: Data are shown as mean ± SD and represent at least two independent experiments with similar results. *$P < 0.05$; **$P < 0.01$; ***$P < 0.001$ (Student's unpaired two-tailed $t$-test). Exact $P$-values are provided in Appendix Table S4.

granulocytic differentiation regulator *C/EBPε* in primary AML specimens, Am80-GCSF effectively inhibits malignant growth while producing well-differentiated functional neutrophils. Moreover, the defined dose range of Am80 when combined with GCSF suggests a "therapeutic window" that can serve as a reference for the future clinical study of Am80-GCSF treatment of CCIN.

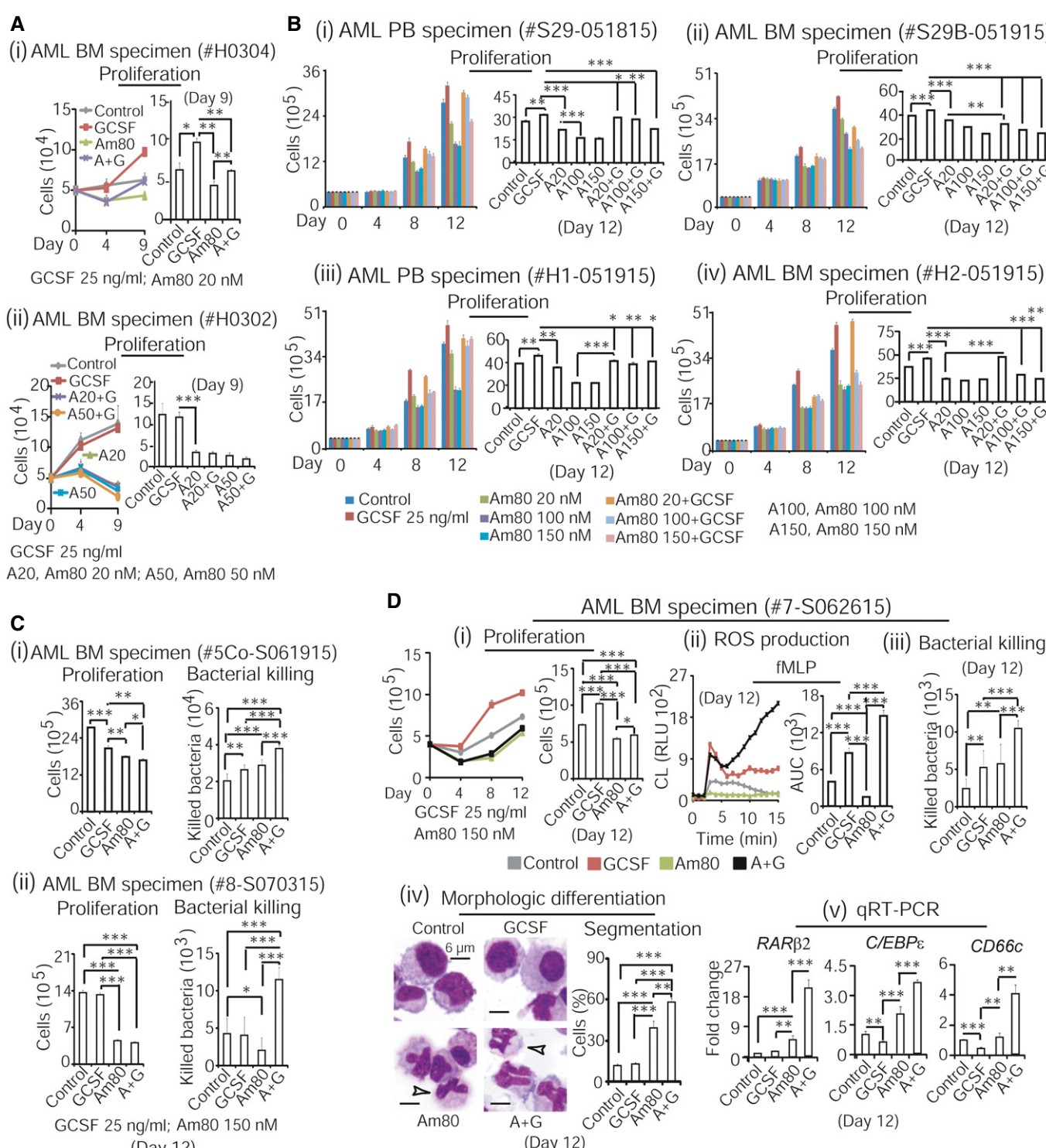

**Figure 3.**

## Am80 alone can effectively differentiate existing granulocytic precursors into functional neutrophils that reduce infection in CCIN mice

To evaluate Am80-GCSF synergy *in vivo*, we first asked whether Am80 could effectively differentiate granulocytic precursors into

functional neutrophils in CCIN mice. Our established mouse CCIN model (Fig 4A), which resembles human CCIN (Appendix Fig S3) described by other groups (Crawford *et al*, 1991; Trillet-Lenoir *et al*, 1993), was applied in the tests. These mouse and human models show that CCIN consists of both neutrophil decrease (granulopoiesis is inhibited) and neutrophil recovery ("emergency"

**Figure 3.  Significant induction of *RAR*β₂ and *C/EBP*ε by Am80-GCSF is associated with both production of functional neutrophils and growth inhibition in primary human leukemic specimens.**

A  Fresh BM mononuclear cells collected from AML patients were treated with 20 or 50 nM Am80 alone or in combination with medium dose of GCSF (25 ng/ml) for up to 9 days. Dynamic changes in proliferation (left sections) and live cells on day 9 (right sections) were illustrated for each of specimens. Controls were without treatment.

B  Fresh PB or BM mononuclear cells collected from AML patients were treated with 20, 100, or 150 nM Am80 alone or in combination with GCSF for up to 12 days. Dynamic changes in proliferation (left sections) and live cells on day 12 (right sections) were illustrated for each of specimens.

C  Fresh BM mononuclear cells collected from AML patients were treated with 150 nM Am80 alone or in combination with GCSF. Proliferation (left sections) and bacterial killing (right sections) were analyzed on day 12 for each of samples.

D  Fresh BM mononuclear cells collected from AML patient were treated with 150 nM Am80 alone or in combination with GCSF. Proliferation was assessed for up to 12 days (i), whereas ROS production, bacterial killing, morphologic differentiation, and RA-target gene expression were assessed at day 12 (ii–v). White arrows indicate neutrophil nuclear segmentation.

Data information: Data are shown as mean ± SD and represent at least two independent experiments with similar results. *$P < 0.05$; **$P < 0.01$; ***$P < 0.001$ (Student's unpaired two-tailed *t*-test). Exact *P*-values are provided in Appendix Table S4.

granulopoiesis) stages, where the risk for microbial infection significantly increases. Because granulopoiesis is suppressed in the neutrophil decrease stage, this provides a unique *in vivo* setting to test whether Am80 could differentiate existing granulocytic precursors into functional neutrophils. We used human equivalent doses (HED) of Am80 and/or GCSF (Fig 4B) in these tests. Since neutrophils induced by high HED of GCSF in CCIN mice failed to fight bacterial infection compared to neutrophils induced by high HED of Am80 (Ding *et al*, 2013), we evaluated low and medium HED of Am80 and/or GCSF. Normal C57BL6/J mice were injected with cancer chemotherapy drug cyclophosphamide (CPA) to induce CCIN, versus control mice without CPA (Appendix Table S3). After 4 h of CPA injection, mice were treated with low or medium doses of Am80 and/or GCSF for 3 days (Fig 4C versus G). On day 2 of neutrophil decrease, Gram-positive bacteria *Staphylococcus aureus* (*S. aureus*) causing bacteremia in neutropenic patients with cancer (Gonzalez-Barca *et al*, 2001) were used to transiently infect mice for up to 16 h. As expected, infection-promoted induction of PB neutrophils only occurred in control mice (Fig 4D and H). Ficoll isolation separated control mice's PB neutrophils from PB mononuclear cells, whereas Percoll gradient segregation of control mice's BM identified that the less and more mature BM neutrophils were concentrated in the second and third layers, respectively (Appendix Fig S4). We found that neutrophil generation was significantly suppressed in both BM and PB of all CCIN mice compared to control mice (Fig 4D, E, H and I). However, among CCIN mice groups, both low and medium doses of Am80 sustained

significantly higher numbers of neutrophils than did GCSF in PB, while having the least in BM compared to GCSF or Am80-GCSF (Fig 4E and I, i versus ii). Interestingly, neutrophil levels maintained by Am80-GCSF were closer to Am80 in PB but more comparable to GCSF in BM (Fig 4E and I). In parallel, PB collected from living mice 3 h post-infection or euthanized mice 16 h post-infection was evaluated for neutrophil bacterial killing. We found that PB neutrophils induced by either low or medium dose of Am80 killed significantly more bacteria than those induced by GCSF or Am80-GCSF (Fig 4F and J). Altogether, because Am80 induces higher numbers of neutrophils in PB while causing the least to remain in BM, this suggests that when granulopoiesis is inhibited in CCIN mice, Am80-enhanced bacterial killing is the result of Am80-induced effective differentiation of existing granulocytic precursors into functional neutrophils.

## Am80-GCSF coordinates myeloid expansion with granulocytic differentiation to generate large amounts of functional neutrophils that reduce infection in CCIN mice

BM "emergency" granulopoiesis for rapidly producing large numbers of neutrophils occurs during the neutrophil recovery stage of mouse or human CCIN (see Fig 4A; Appendix Fig S3). We addressed whether, during this stage, the potent effect of Am80 on neutrophil differentiation (see Fig 4) could synergize with GCSF-mediated myeloid expansion to generate sufficient numbers of functional neutrophils fighting infection. Because either low or medium

**Figure 4.  Am80 alone can effectively differentiate existing granulocytic precursors into functional neutrophils that reduce infection in CCIN mice.**

A  Mouse CCIN induced by cancer chemotherapy drug cyclophosphamide (CPA) consists of both neutrophil decrease and neutrophil recovery stages. ANC, absolute neutrophil count.

B  Equivalent low, medium, and high doses of Am80 and GCSF between human and mouse were calculated following FDA dose conversion guidelines.

C  Neutrophil decrease model with low-dose treatment. After 4 h of CPA injection, mice were treated with regimens for 3 days. On day 2, mice were infected with $6 \times 10^6$ CFU of *S. aureus* via intravenous injection for up to 16 h before euthanasia. Control mice without CPA. N, numbers of mice.

D  Vetscan counting PB leukocytes.

E  Neutrophils induced by low-dose treatment in PB (i) and BM (ii).

F  Bacterial killing by PB neutrophils was assessed at 3 and 16 h post-infection and in spleen (i–iii), using blood agar analysis of total extracellular viable bacteria.

G  Neutrophil decrease model with medium-dose treatment, using similar procedures described in panel (C).

H  Vetscan counting PB leukocytes.

I  Neutrophils induced by medium-dose treatment in PB (i) and BM (ii).

J  Similar to panel (F), bacterial killing by PB neutrophils was evaluated 3 and 16 h post-infection and in heart.

Data information: Data are shown as mean ± SD. These data represent: 1) two independent low dose tests with similar results; and 2) one time of low and medium dose test performed in parallel. *$P < 0.05$; **$P < 0.01$; ***$P < 0.001$ (Student's unpaired two-tailed *t*-test). Exact *P*-values are provided in Appendix Table S4.

dose of Am80 effectively differentiated granulocytic precursors into functional neutrophils (see Fig 4), we used the low-dose Am80-GCSF in this test. As illustrated (Fig 5A), 2 days after CPA injection,

mice were treated with low-dose regimens for 3 days. Then, mice were transiently infected with *S. aureus* via tail vein for up to 16 h. As expected, significant increases in PB neutrophils were observed

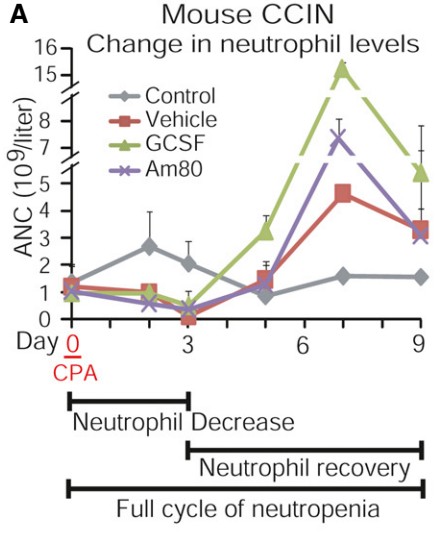

**A**  Mouse CCIN
Change in neutrophil levels

- Control
- Vehicle
- GCSF
- Am80

**B**  Equivalent doses between mouse and human

| Am80 (oral, daily) | Mouse (mg/kg) | Human (mg/m²) | | GCSF (s.c., daily) | Mouse (µg/kg) | Human (µg/kg) |
|---|---|---|---|---|---|---|
| Low dose | 0.5 | 1.5 | | Low dose | 25 | 2 |
| Medium dose | 1 | 3 | | Medium dose | 50 | 4 |
| High dose | 5 | 15 | | High dose | 250 | 20 |

Human dose of Am80 for APL: 6 to 9 mg/m²/day for up to 56 days.
Human dose of GCSF for CCIN: 5 to 10 µg/kg/day for up to 14 days; s.c.: subcutaneous.

**C**  Low Doses
Neutrophil decrease model (N = 19)

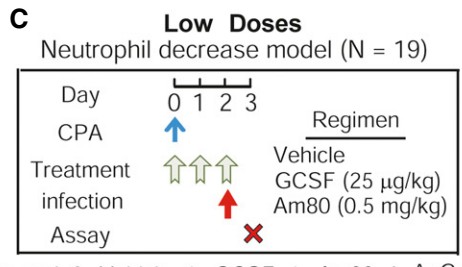

Control: 3;  Vehicle: 4;  GCSF: 4;  Am80: 4;  A+G: 4

**D**  Leukocyte levels

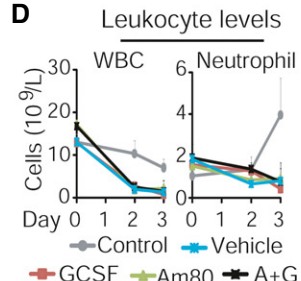

- Control
- Vehicle
- GCSF
- Am80
- A+G

**E**

(i) PB neutrophils (day 3)

| Group | Total cells 1x10⁴/ml | Neutrophils 1x10⁴/ml | (%) |
|---|---|---|---|
| Control | 177 | 134 | 75 |
| Vehicle | 35 | 13 | 38 |
| GCSF | 22 | 7 | 33 |
| Am80 | 22 | 11*** | 49 |
| A+G | 32 | 17 | 52 |

(ii) BM neutrophils (3rd layer; day 3)

| Group | Total cells 1x10⁴/ml | Neutrophils 1x10⁴/femur | (%) |
|---|---|---|---|
| Control | 380 | 250 | 66 |
| Vehicle | 24 | 1 | 6 |
| GCSF | 52 | 8 | 16 |
| Am80 | 16 | 2*** | 10 |
| A+G | 82 | 15 | 18 |

***: Am80 vs. GCSF

**F**  Bacterial killing

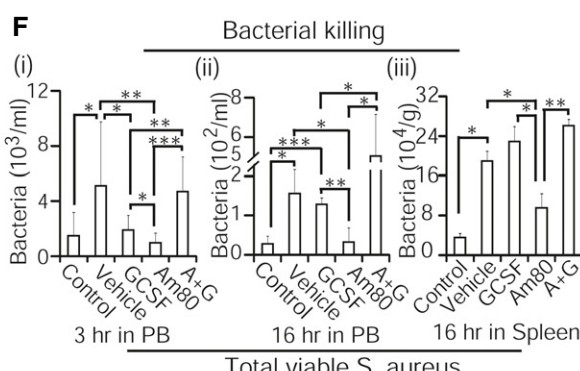

(i) 3 hr in PB    (ii) 16 hr in PB    (iii) 16 hr in Spleen
Total viable *S. aureus*

**G**  Medium Doses
Neutrophil decrease model (N = 19)

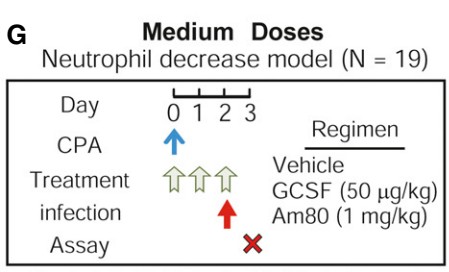
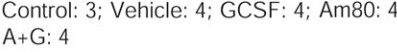

Control: 3;  Vehicle: 4;  GCSF: 4;  Am80: 4; A+G: 4

**H**  Leukocyte levels

- Control
- Vehicle
- GCSF
- Am80
- A+G

**J**  Bacterial killing

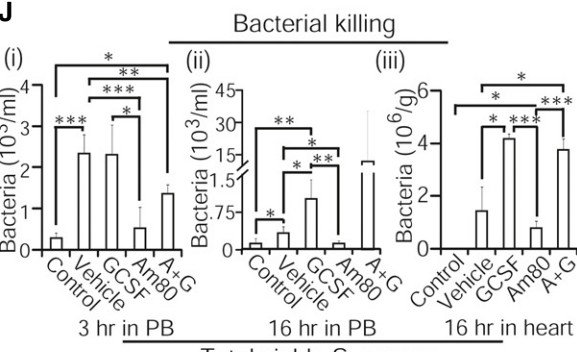

(i) 3 hr in PB    (ii) 16 hr in PB    (iii) 16 hr in heart
Total viable *S. aureus*

**I**

(i) PB neutrophils (day 3)

| Group | Total cells 1x10⁴/ml | Neutrophils 1x10⁴/ml | (%) |
|---|---|---|---|
| Control | 54 | 36 | 67 |
| Vehicle | 9 | 6 | 67 |
| GCSF | 10 | 2** | 17 |
| Am80 | 17 | 7 | 43 |
| A+G | 21 | 10 | 47 |

(ii) BM neutrophils (3rd layer; day 3)

| Group | Total cells 1x10⁴/ml | Neutrophils 1x10⁴/femur | (%) |
|---|---|---|---|
| Control | 184 | 110 | 60 |
| Vehicle | 7 | 2 | 24 |
| GCSF | 21 | 7* | 33 |
| Am80 | 19 | 2 | 9 |
| A+G | 45 | 10 | 23 |

**/*: GCSF vs. all other groups

**Figure 4.**

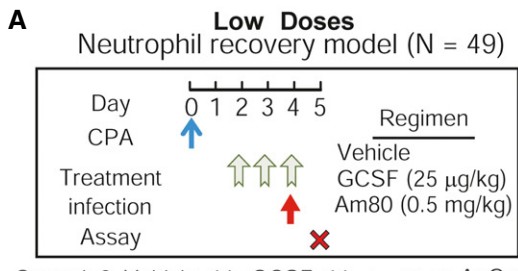

**A**

**Low Doses**
**Neutrophil recovery model (N = 49)**

Day: CPA (day 0)
Treatment infection: days 2,3,4
Assay: day 5

Regimen: Vehicle; GCSF (25 µg/kg); Am80 (0.5 mg/kg)

Control: 8; Vehicle: 11; GCSF: 11; Am80: 8; A+G: 11

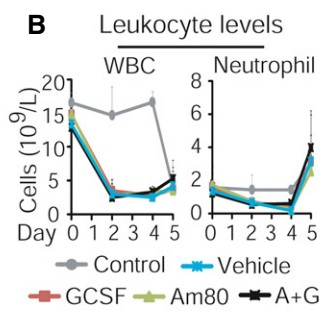

**B** Leukocyte levels

WBC / Neutrophil

Control — Vehicle — GCSF — Am80 — A+G

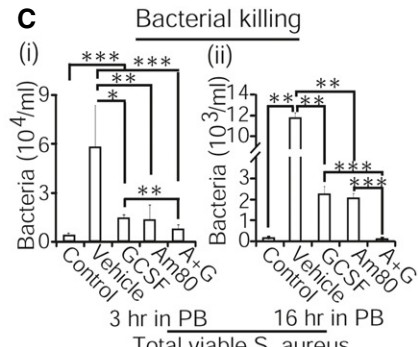

**C** Bacterial killing

(i) 3 hr in PB    (ii) 16 hr in PB

Total viable *S. aureus*

**D** BM neutrophil recovery (day 5)

**Second layer**

| Group | Total cells $1\times10^4$/femur | Neutrophils $1\times10^4$/femur | (%) |
|---|---|---|---|
| Control | 12 | 4 | 33 |
| Vehicle | 42 | 7 | 17 |
| GCSF | 129 | 71 | 55 |
| Am80 | 84 | 23** | 27 |
| A+G | 188 | 73 | 39 |

**: Am80 vs. A+G and GCSF

**Third layer**

| Group | Total cells $1\times10^4$/femur | Neutrophils $1\times10^4$/femur | (%) |
|---|---|---|---|
| Control | 90 | 57 | 63 |
| Vehicle | 39 | 8 | 20 |
| GCSF | 93 | 45 | 48 |
| Am80 | 49 | 10** | 20 |
| A+G | 46 | 26 | 57 |

**: Am80 vs. A+G and GCSF

Granulocytic morphologic differentiation

**Second layer**

Control | Vehicle | GCSF | Am80 | A+G

6 µm

**Third layer**

Control | Vehicle | GCSF | Am80 | A+G

6 µm

**E** PB neutrophil recovery (day 5)

**Mononuclear layer**

| Group | Total cells $1\times10^4$/ml | Neutrophils $1\times10^4$/ml | (%) |
|---|---|---|---|
| Control | 11 | 4 | 36 |
| Vehicle | 23 | 11 | 48 |
| GCSF | 20 | 14 | 70 |
| Am80 | 13 | 6 | 46 |
| A+G | 43 | 29*** | 67 |

***: A+G vs. all other groups

**Neutrophil layer**

| Group | Total cells $1\times10^4$/ml | Neutrophils $1\times10^4$/ml | (%) |
|---|---|---|---|
| Control | 82 | 62*** | 76 |
| Vehicle | 11 | 3 | 27 |
| GCSF | 45 | 33 | 73 |
| Am80 | 45 | 15** | 27 |
| A+G | 45 | 32 | 71 |

**: Am80 vs. A+G and GCSF
***: Control vs. all other groups

Granulocytic morphologic differentiation

**Mononuclear layer**

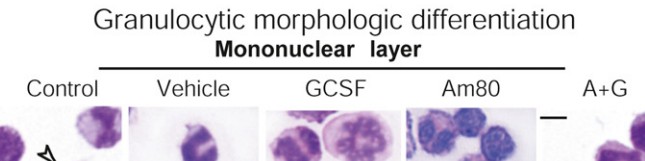

Control 1 | Vehicle 2 | GCSF 3 | Am80 4 | A+G 5

6 µm

**Neutrophil layer**

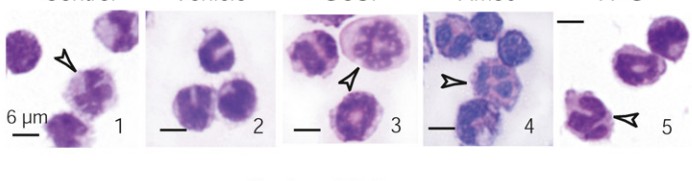

Control 1 | Vehicle 2 | GCSF 3 | Am80 4 | A+G 5

6 µm

**Figure 5.**

in all groups after bacterial infection (Fig 5B). PB collected at 3 and 16 h post-infection was used to assess neutrophil bactericidal activity. All three independent experiments showed that neutrophils generated by Am80-GCSF, but not by Am80 or GCSF alone, killed significantly more bacteria (Fig 5C). By evaluating neutrophil generation in these CCIN mice, we found that in either the BM's second

**Figure 5.  Am80-GCSF coordinates myeloid expansion with granulocytic differentiation to generate large amounts of functional neutrophils that reduce infection in CCIN mice.**

A   After 48 h of CPA injection, mice were treated with low doses of Am80 and/or GCSF for 3 days. Mice were infected with $9 \times 10^6$ CFU of *S. aureus* through intravenous injection on day 4 and sacrificed 16 h post-infection. Control mice without CPA.

B   Vetscan counting PB leukocytes.

C   Bacterial killing by PB neutrophils was evaluated 3 and 16 h post-infection (i, ii), using blood agar analysis of extracellular viable bacteria.

D, E   Neutrophil recovery and morphologic differentiation induced by low doses of Am80 and/or GCSF in BM (D) and PB (E). White arrows indicate neutrophil nuclear segmentation.

Data information: Data are shown as mean ± SD and represent three independent experiments with similar results. *$P < 0.05$; **$P < 0.01$; ***$P < 0.001$ (Student's unpaired two-tailed *t*-test). Exact *P*-values are provided in Appendix Table S4.

layer (containing less mature neutrophils) or third layer (containing more mature neutrophils) or the PB's mononuclear or neutrophil layer, Am80 induced fewer neutrophils compared to GCSF or Am80-GCSF (Fig 5D and E; left sections). Both GCSF and Am80-GCSF induced significantly greater numbers of morphologically differentiated neutrophils in PB (Fig 5E). However, in both PB mononuclear and neutrophil layers, the degree of neutrophil nuclear segmentation induced by Am80-GCSF was similar to those in control mice but significantly higher than those in GCSF or Am80 mice (Fig 5E, right sections, images 5 versus 1 or 3 or 4). These results indicate that in the neutrophil recovery stage of mouse CCIN, Am80 has the least capacity to promote BM granulopoiesis, whereas GCSF induces large amounts of immature neutrophils as shown by both bacterial killing and neutrophil nuclear segmentation (Fig 5C–E). However, Am80-GCSF can synergize Am80's ability of granulocytic differentiation with GCSF's competence of myeloid expansion, thus generating large amounts of functional neutrophils that reduce infection in CCIN mice.

## Am80-GCSF reduces infection-related mortality in CCIN mice undergoing perpetual systemic intravenous bacterial infection throughout a full cycle of mouse CCIN

Since transient bacterial infection was effectively reduced in CCIN mice treated with Am80-GCSF (see Fig 5), we further addressed whether this Am80-GCSF synergy could reduce infection-related mortality of CCIN mice undergoing perpetual systemic intravenous bacterial infection, using three different dose-schedule-survival models covering a full cycle of mouse CCIN. As illustrated in survival model #1 (Fig 6A), mice were given low doses of Am80 and/or GCSF after 4 h of CPA injection, and *S. aureus* infection was initiated on day 3. Two survival measurements, including general survival on day 9 following CPA injection and infectious survival after 6 days of *S. aureus* infection (Fig 6Bi and ii), demonstrated that Am80-GCSF significantly reduced infection-related mortality in CCIN mice. Three independent experiments showed similar results (Appendix Fig S5A). To examine bactericidal functions of newly generated neutrophils, mice surviving on day 9 from control and Am80-GCSF groups of survival model #1 were again injected with *S. aureus* or vehicle solution for 15 min before euthanasia. Parallel Giemsa and Gram stains showed that PB neutrophils from infected mice were able to phagocytose bacteria, whereas PB neutrophils from mice injected with vehicle solution were free of bacteria (Fig 6C). Since GCSF is usually given to patients ≥ 24 h post-administration of chemotherapy drug, we tested both low and medium doses of Am80-GCSF treatment 24 h post-CPA, as illustrated

(Fig 6D and F). Two independent tests were performed for each dose in these survival models, respectively (Appendix Fig S5B and C). We found that with low-dose treatment, Am80-GCSF mice experienced 100% survival compared to only 12.5% survival in GCSF mice (Fig 6E). In contrast, medium-dose treatment showed no statistical difference in survival rate between Am80-GCSF (55.6%) and GCSF (11.1%) groups (Fig 6G). Altogether, these data demonstrate that low-dose Am80-GCSF can effectively reduce infection-related mortality of CCIN mice experiencing perpetual bacterial infection.

## Am80-GCSF generates sufficient numbers of functional neutrophils while preventing myeloid overexpansion during "emergency" granulopoiesis in CCIN mice

Further, we investigated *in vivo* myeloid expansion and granulocytic differentiation mediated by Am80-GCSF. By examining myeloid production versus neutrophil differentiation in CCIN mice surviving on day 9, we found that GCSF-induced BM myeloid cells concentrated in the second layer, known to contain less mature neutrophils (Fig 7A, upper section). In contrast, more mature BM neutrophils known to localize in the third layer were significantly induced by Am80-GCSF (Fig 7A, lower section). GCSF mice produced the largest amount of morphologically differentiated PB neutrophils compared to control or Am80-GCSF mice (Fig 7B). However, these neutrophils displayed less intense nuclear segmentation (Fig 7B, right section, neutrophil layer) and, importantly, failed to reduce infection and infection-related mortality in CCIN mice (see Figs 5 and 6). Remarkably, the increased production of PB neutrophils in Am80-GCSF mice during "emergency" granulopoiesis (Fig 7B) not only returned to the similar levels shown in control mice on day 13, but also displayed nuclear segmentation similar to those in control mice (Fig 7Ci and ii). Moreover, the increased spleen size and weight in Am80-GCSF mice on day 9 dropped to the levels similar to those in control mice on day 13 (Fig 7Ciii and iv). Compared to GCSF mice, neutrophil production in Am80-GCSF mice was not associated with significant loss of body weight (Fig 7D). Further ultrastructural analysis of newly generated PB neutrophils collected from mice surviving on day 11 showed that in clear contrast to neutrophils from control or Am80-GCSF mice, neutrophils from GCSF mice displayed cellular degeneration. This was evidenced by significantly increased numbers of variably sized intracytoplasmic vacuoles, displaying empty or homogeneous matrix and irregular amorphous material or myelin structure (Fig 7Ei and ii). Neutrophils from GCSF mice also displayed markedly higher levels of inner nuclear membrane dilation or separation from outer nuclear membrane, an indication of disrupted nuclear envelope (Fig 7Ei,

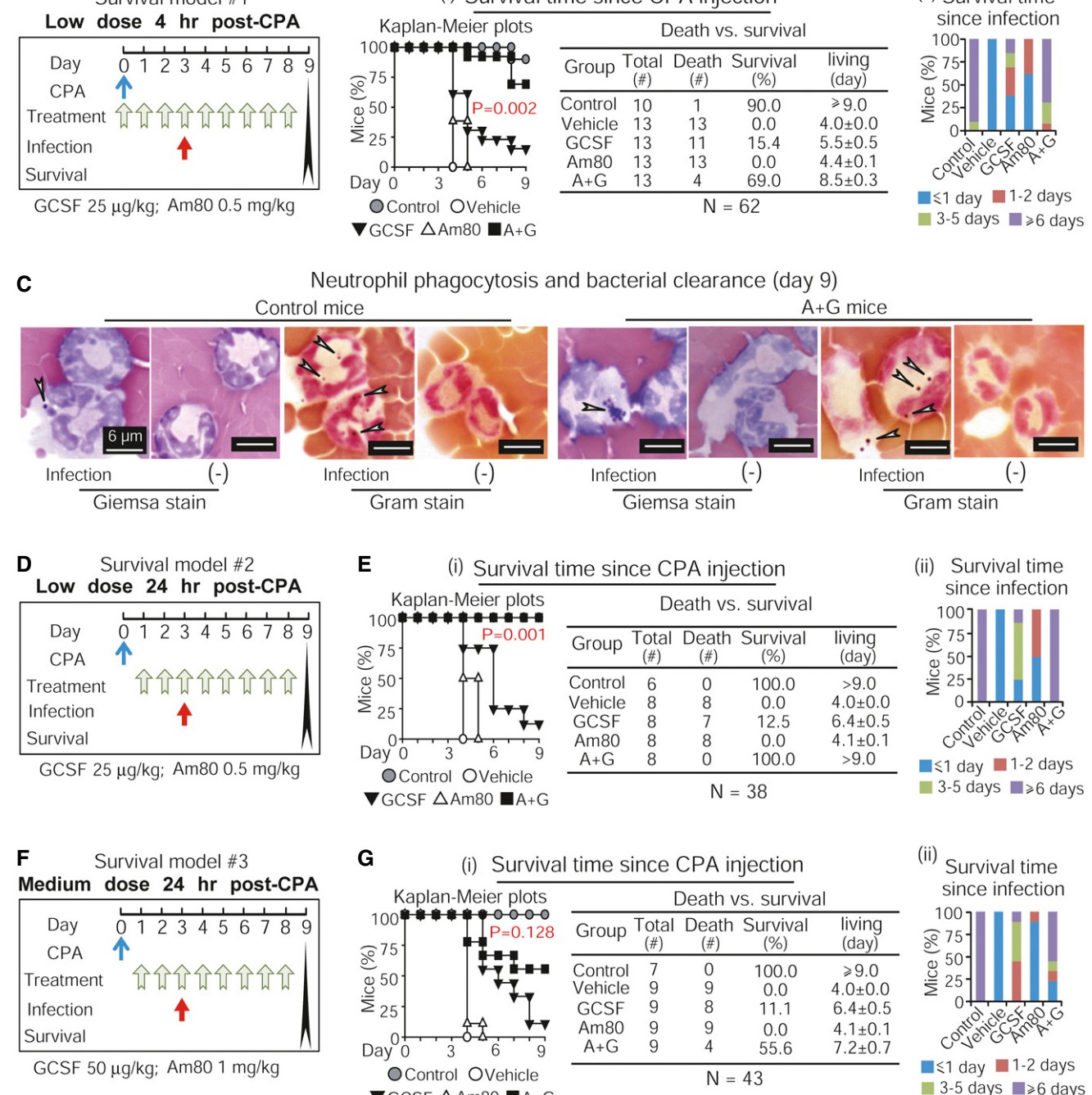

Figure 6.  Am80-GCSF reduces infection-related mortality in CCIN mice undergoing perpetual systemic intravenous bacterial infection throughout a full cycle of mouse CCIN.

A    Survival model #1 with low-dose treatment after 4 h of CPA injection. Mice were infected with $5 \times 10^6$ CFU of *S. aureus* via intravenous injection on day 3. Moribund mice were euthanized and recorded as deceased on the day of euthanasia. Control mice without CPA.

B    Survival of CCIN mice was calculated with both Kaplan–Meier plots and log-rank test on day 9 since CPA injection (i) as well as assessed 6 days post-infection (ii). Data represent three independent experiments with similar results.

C    Gram and Giemsa stains analyzed neutrophil phagocytosis and bacterial clearance in mice surviving on day 9. White arrows indicate phagocytosed *S. aureus* in neutrophils.

D    Survival model #2 with low-dose treatment after 24 h of CPA injection. Other schedules/procedures were the same to those described in (A).

E    Survival of CCIN mice was calculated with both Kaplan–Meier plots and log-rank test on day 9 since CPA injection (i) as well as assessed 6 days post-infection (ii). Data represent two independent experiments with similar results.

F    Survival model #3 with medium-dose treatment after 24 h of CPA injection. Other schedules/procedures were the same to those described in (A).

G    Survival of CCIN mice was calculated with both Kaplan–Meier plots and log-rank test on day 9 since CPA injection (i) as well as assessed 6 days post-infection (ii). Data represent two independent experiments with similar results.

## A    BM neutrophil recovery (day 9)

### Second layer

| Group | Total cells 1x10$^4$/femur | Neutrophils 1x10$^4$/femur | (%) |
|---|---|---|---|
| Control | 467 | 89 | 19 |
| GCSF | 683 | 327** | 48 |
| A+G | 407 | 196* | 48 |

*/**: vs. Control

### Third layer

| Group | Total cells 1x10$^4$/femur | Neutrophils 1x10$^4$/femur | (%) |
|---|---|---|---|
| Control | 282 | 175 | 62 |
| GCSF | 661 | 234 | 35 |
| A+G | 680 | 575*** | 85 |

***: A+G vs. all other groups

### Morphologic differentiation

**Second layer**
Control    GCSF    A+G
6 µm

**Third layer**
Control    GCSF    A+G
1    2    3
6 µm

## B    PB neutrophil recovery (day 9)

### Mononuclear layer

| Group | Total cells 1x10$^4$/ml | Neutrophils 1x10$^4$/ml | (%) |
|---|---|---|---|
| Control | 247 | 79** | 32 |
| GCSF | 968 | 509 | 53 |
| A+G | 826 | 386 | 47 |

**: Control vs. all other groups

### Neutrophil layer

| Group | Total cells 1x10$^4$/ml | Neutrophils 1x10$^4$/ml | (%) |
|---|---|---|---|
| Control | 338 | 234*** | 69 |
| GCSF | 1,410 | 1,187 | 84 |
| A+G | 1,062 | 877 | 83 |

***: Control vs. all other groups

### Morphologic differentiation

**Mononuclear layer**
Control    GCSF    A+G
6 µm

**Neutrophil layer**
Control    GCSF    A+G
1    2    3
6 µm

## C    (i) PB neutrophil recovery (day 13)

### Mononuclear layer

| Group | Total cells 1x10$^4$/ml | Neutrophils 1x10$^4$/ml | (%) |
|---|---|---|---|
| Control | 541 | 130** | 24 |
| A+G | 504 | 280 | 55 |

### Neutrophil layer

| Group | Total cells 1x10$^4$/ml | Neutrophils 1x10$^4$/ml | (%) |
|---|---|---|---|
| Control | 494 | 457 | 93 |
| A+G | 453 | 431 | 95 |

(ii) Morphologic differentiation
**Mononuclear layer**    **Neutrophil layer**
Control    A+G    Control    A+G
6 µm

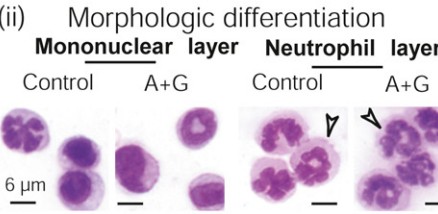

(iii) Spleen size
day 9    day 13
Control
GCSF    No surviving GCSF mice on day 13
A+G
— 0.5 cm

(iv) Spleen weight
Control
GCSF
A+G
Weight (Gram) 0.4 0.3 0.2 0.1 0
Day 0 3 6 9 12 15
*

## D    Body weight

Control
GCSF
A+G
Weight (Gram) 24 21 18 15 12 9
0 3 6 9 (Day)
*

## E    Neutrophil ultrastructures
(Day 11)

(i)
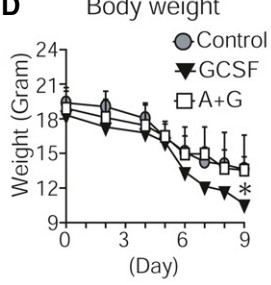
GCSF    Sg    Pg    1    1 µm

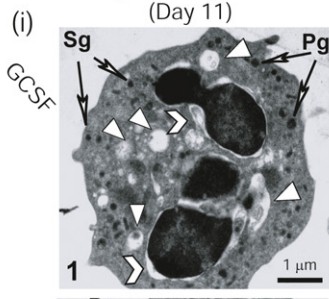
A+G    Pg    Sg    2

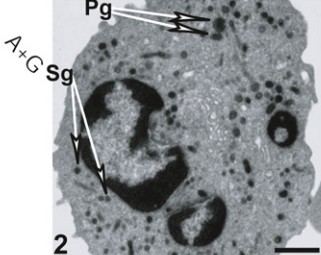
Control    Pg    Sg    Pg    3

▷ Vacuoles    ➤ Granules
⧩ Nuclear membrane dilation

(ii) Vacuoles
Cells (%) 90 60 30 0
Control A+G GCSF
**

(iii) Membrane dilation
Cells (%) 40 30 20 10 0
Control A+G GCSF
**

Figure 7.

images 1 versus 2 or 3; 7Eiii). Furthermore, less numbers of granules appeared in neutrophils from GCSF mice than in the neutrophils of control or Am80-GCSF mice (Fig 7Ei). Altogether, these results demonstrate that during "emergency" granulopoiesis of mouse CCIN, GCSF produces the largest amount of immature neutrophils. In contrast, Am80-GCSF synergizes myeloid expansion

◄

**Figure 7.  Am80-GCSF prevents myeloid overexpansion while generating sufficient numbers of functional neutrophils during "emergency" granulopoiesis in CCIN mice.**

A, B  Neutrophil recovery and morphologic differentiation were assessed in BM (A) and PB (B) of CCIN mice surviving on day 9. White arrows indicate neutrophil nuclear segmentation.

C    PB neutrophil production and nuclear segmentation were assessed in control and Am80-GCSF mice surviving on day 13 (i, ii).  White arrows indicate neutrophil nuclear segmentation. The increased spleen size and weight on day 9 in Am80-GCSF mice dropped to the levels similar to those in control mice on day 13 (iii, iv).

D    Significant loss of body weight occurred in GCSF mice on day 9.

E    Ultrastructure analysis of neutrophils collected from mice surviving on day 11. Neutrophils from GCSF mice showed less granules (i, images 1 versus 2 or 3), more intracytoplasmic vacuoles (i, ii), and inner nuclear membrane dilation (i, iii). Pg: primary granules; Sg: secondary granules.

Data information: Data are shown as mean ± SD and represent at least two independent experiments with similar results. *$P < 0.05$; **$P < 0.01$; ***$P < 0.001$ (Student's unpaired two-tailed *t*-test). Exact *P*-values are provided in Appendix Table S4.

with effective granulocytic differentiation to generate sufficient numbers of functional neutrophils without causing myeloid overexpansion.

## Discussion

The numbers of functional neutrophils induced by Am80 are limited (Ding *et al*, 2013). Although GCSF stimulates profound myeloid expansion, it cannot effectively differentiate those precursors into functional neutrophils fighting infection (Leavey *et al*, 1998; Donini *et al*, 2007; Dick *et al*, 2008; Ding *et al*, 2013). We therefore addressed, in this study, whether and how Am80 can synergistically work with GCSF during granulopoiesis to generate sufficient numbers of functional neutrophils while preventing myeloid overexpansion. We find that Am80-GCSF sustains active proliferation in normal primary human hematopoietic precursors while inhibiting leukemic growth in primary human AML specimens. However, it ultimately generates functional neutrophils in these different cell types compared to Am80 or GCSF. These functional neutrophils display significantly increased ROS production, enhanced bacterial killing, and altered transcription of RA-target genes that promote granulocytic differentiation to develop neutrophil innate immunity (Figs 1–3; Appendix Figs S1 and S2). Further *in vivo* evaluations demonstrate that large amounts of GCSF-induced neutrophils are functionally immature, incapable of reducing mouse CCIN-associated infection and mortality (Figs 4–7). In contrast, although Am80 shows little capacity to promote BM myeloid expansion, the potent effect of Am80 on differentiating granulocytic precursors in CCIN mice (Fig 4) can synergize with GCSF-dependent myeloid expansion, resulting in the generation of large amounts of functional neutrophils that effectively reduce infection in CCIN mice (Fig 5). Convincingly, different survival tests demonstrate that such Am80-GCSF synergy, without causing myeloid overexpansion, generates sufficient numbers of functional neutrophils to reduce infection-related mortality in CCIN mice undergoing perpetual systemic intravenous bacterial infection during a full cycle of mouse CCIN (Figs 6 and 7). These findings reveal a novel Am80-GCSF synergy, suggesting an advanced therapy that has potential efficacy for CCIN treatment.

RA-RARα signaling is essential for transcriptional control of granulocytic differentiation (Melnick & Licht, 1999; Collins, 2008). Am80-GCSF selectively modulates RARα action to alter transcription of RA-target genes that promote the development of neutrophil innate immunity (Figs 1–3; Appendix Figs S1 and S2). Accordingly, neutralizing CR3-dependent neutrophil innate immunity with specific antibodies eliminates neutrophil bactericidal activities induced by Am80-GCSF (Fig 1). How does Am80-GCSF enhance such

immunity development during myeloid expansion? GCSF accelerates neutrophil production and induces mobilization and expansion of HSC (Panopoulos & Watowich, 2008). Unfortunately, GCSF is not effective in mediating neutrophil differentiation to develop innate immunity (Figs 1–7) (Leavey *et al*, 1998; Donini *et al*, 2007; Dick *et al*, 2008; Ding *et al*, 2013). It is known that the C/EBP family of transcription factors is important in regulating GCSF-dependent myeloid expansion and granulocytic differentiation (Panopoulos & Watowich, 2008). Among them, GCSF-induced *C/EBP*β is required for GCSF-mediated expansion of the granulocytic compartment *in vivo* (Hirai *et al*, 2006), whereas *C/EBP*ε induced by GCSF via C/EBPα-dependent or C/EBPα-independent pathways (Nakajima & Ihle, 2001; Numata *et al*, 2005; Panopoulos & Watowich, 2008) is involved in mediating GCSF-dependent neutrophil differentiation. Conversely, RARα transactivates *C/EBP*ε to regulate terminal granulocytic differentiation (Park *et al*, 1999; Lekstrom-Himes, 2001) while mediating expression of *CD66*, *CD18*, and *CD11b* (Park *et al*, 1999; Bush *et al*, 2003) that are essential for the development of neutrophil innate immunity (Groves *et al*, 2008). Additionally, Am80 is more potent than RA in transcription of *C/EBP*ε, *C/EBP*β, *CD66c*, and *CD18* (Appendix Fig S1D). Hence, in balancing GCSF induction of *C/EBP*β on promoting myeloid expansion, Am80-GCSF may induce adequate "emergency" granulopoiesis by altering RA-dependent transcription of *C/EBP*ε, *CD66*, *CD18,* and *CD11b* to promote differentiation of granulocytic precursors into functional neutrophils (Fig 8).

Granulopoiesis demands coordinated proliferation and differentiation of granulocytic progenitors. In normal primary human hematopoietic precursors, Am80-GCSF synergizes active proliferation with effective granulocytic differentiation to generate significantly larger amounts of functional neutrophils than does Am80 (Fig 1). In contrast, Am80-GCSF inhibits malignant growth while still producing functional neutrophils in primary human AML specimens (Fig 3). How can Am80-GCSF modulate such differential processes in normal versus malignant cells? It is known that RARβ$_2$ is a master tumor suppressor (Soprano *et al*, 2004; Alvarez *et al*, 2007) whereas C/EBPε is a key regulator of terminal granulocytic differentiation (Park *et al*, 1999; Lekstrom-Himes, 2001). Both *RAR*β$_2$ and *C/EBP*ε are the direct transcriptional targets of RARα (de The *et al*, 1990; Melnick & Licht, 1999; Park *et al*, 1999; Lekstrom-Himes, 2001). Notably, Am80-GCSF induces significantly higher transcription of both *RAR*β$_2$ and *C/EBP*ε than does Am80 in the early and late differentiation induction stages in either normal or malignant cells (Figs 1–3). On the other hand, whereas both GCSF and Am80-GCSF induce significantly higher expression of *RAR*β$_2$ and *C/EBP*ε than does Am80 in the late differentiation induction stage of

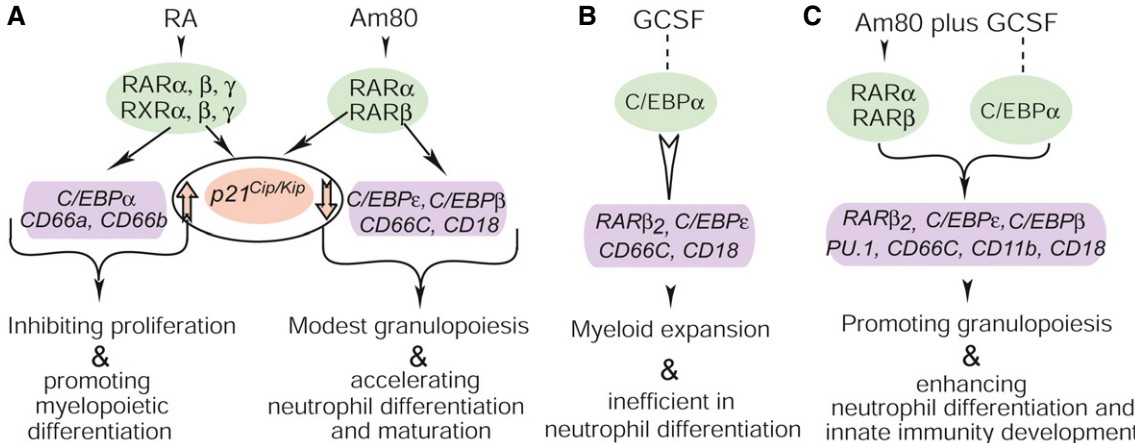

**Figure 8. Am80-GCSF synergizes myeloid expansion with granulocytic differentiation by altering transcription of RA-target genes.**

A   Am80 selectively activates RARα and RARβ to alter expression of RA-target genes, leading to modest myeloid expansion and effective neutrophil differentiation. Up arrow, increased expression of cell cycle inhibitor $p21^{Cip/Kip}$; down arrow, decreased expression of $p21^{Cip/Kip}$.

B   GCSF has limited effect on mediating transcription of RA-target genes that promote granulocytic differentiation. Dot line, non-direct effect on gene transcription; white arrow, limited effect on gene transcription.

C   Am80-modulated selective activation of RARα and RARβ synergizes with GCSF induction of C/EBPα-dependent and C/EBPα-independent transcriptional regulation, resulting in effective differentiation of large amounts of granulocytic precursors into functional neutrophils during myeloid expansion.

normal cells, only Am80-GCSF consistently induces significantly higher expression of both $RAR\beta_2$ and $C/EBP\varepsilon$ than does GCSF throughout the full differentiation induction period in primary human AML specimens (Figs 1B versus 2Bi and 3Dv). Thus, Am80-GCSF may differentially modulate expression of $RAR\beta_2$ and $C/EBP\varepsilon$ in normal versus malignant cells to synergize granulocytic differentiation with myeloid expansion. Additionally, RARα is a substrate for cyclin-dependent kinase-activating kinase (CAK) complex (Rochette-Egly *et al*, 1997) that cross-regulates cell cycle with transcription (Rochette-Egly *et al*, 1997; Fuss & Tainer, 2011). RA decreases CAK phosphorylation of RARα, leading to induction of granulocytic differentiation in both normal and malignant human hematopoietic precursors (Luo *et al*, 2007; Wang *et al*, 2010; Lou *et al*, 2013). Hence, Am80-GCSF may also induce changes in RARα phosphorylation to modulate both transcription and cell cycle to coordinate myeloid expansion with neutrophil differentiation. By mapping the resultant transactivation network of RARα induced by Am80-GCSF under the modulation of RNAi silencing or gene targeting in normal versus malignant hematopoietic progenitors, future studies should define an array of transcription factors that coordinate with RARα at distinct developmental stages to differentially regulate production of functional neutrophils.

## Materials and Methods

### Primary human specimens and culture conditions

Approval for use of primary human specimens was obtained from Children's Hospital Los Angeles (CHLA) Institutional Review Board. Informed consent was obtained from all individuals and the study was conducted in accordance with the Declaration of Helsinki. Fresh BM and PB specimens were collected from normal human donors and non-APL AML patients. Lineage-specific

myeloid medium plus 30% human stromal cell-conditioned medium (MM-HSCCM), as described (Luo *et al*, 2007; Chaudhry *et al*, 2012; Ding *et al*, 2013; Lou *et al*, 2013), was used for culturing primary human specimens. In brief, Iscove's modified Dulbecco's medium (Gibco-BRL, Gaithersburg, MD) was mixed with 20% fetal calf serum (FCS), 1% bovine serum albumin (Sigma-Aldrich, St. Louis, MO), $10^{-4}$ mol/l 2-mercaptoethanol (Sigma-Aldrich), $10^{-6}$ mol/l hydrocortisone (Sigma-Aldrich), 292 mg/l glutamine (Corning, Manassas, VA), and a combination of 10 ng/ml interleukin-3 (IL-3; R&D, McKinley Place NE, MN), 2.5 ng/ml IL-6 (R&D), 25 ng/ml stem cell factor (R&D), and 30% human stromal cell-conditioned medium. The MM-HSCCM sustains myelopoiesis while blocking the growth of lymphoid and erythroid cells in the presence of hydrocortisone and in the absence of erythropoietin. Primary cells were split every 3 days with equal portions of fresh MM-HSCCM.

### Normal primary human hematopoietic precursors and cell culture

Human umbilical cord blood CD34$^+$ precursors (AllCells, Emeryville, CA) were expanded, as described (Chaudhry *et al*, 2012). Culture conditions for myelopoiesis of expanded CD34$^+$ precursors, as described (Chaudhry *et al*, 2012), were the same to those for culturing primary human specimens.

### Isolations of human neutrophils and mononuclear cells

Fresh BM and PB specimens of normal human donors or non-APL AML patients were collected in heparin-coated Vacutainer coagulation tubes (BD Biosciences, San Jose, CA), diluted with one volume of Hank's balanced salt solution (HBSS), loaded onto half volume of Ficoll–Paque Premium 1.077 (GE Healthcare, Piscataway, NJ), and centrifuged at 400 × g for 40 min at room temperature (RT). The

mononuclear cells collected from the upper layer of the Ficoll were further mixed with three volumes of HBSS and centrifuged at $500 \times g$ for 10 min. Erythrocytes contaminated in neutrophils or mononuclear cells during Ficoll separation were removed by mixing cells with 0.5 ml sterile water for 10 s, and cell mixture was then immediately diluted with 40 ml HBSS to minimize hypotonic damage. Cells were collected with low-speed centrifugation with $500 \times g$ for 5 min at 4°C.

## Converting human plasma concentrations of GCSF and Am80 to equivalent cell culture doses for *in vitro* tests

GCSF (Neupogen, Amgen Mfg. Ltd) was from CHLA Pharmacy, and Am80 was provided by Dr. Shudo. The established dose of 25 ng/ml GCSF for inducing granulocytic differentiation (Luo *et al*, 2007; Ding *et al*, 2013; Lou *et al*, 2013) as well as different Am80 doses was calculated, respectively (see Appendix Table S1), based on their human plasma concentration profiles derived from pharmacokinetics (PK) studies using human subjects [NEUPOGEN® (filgrastim), www.neupogen.com, Prescribing Information, 1991–2015 Amgen, Inc.; Investigator's brochure, tamibarotene (INNO-507)]. For example, the peak plasma concentrations ($C_{max}$) derived from the PK of Am80 oral doses 2, 4, and 6 mg/m$^2$ are equivalent to 51, 128, and 171 nM, respectively. The equation is as follows: $C_{max}$ (ng/ml) $\times$ 1,000 = $C_{max}$ (nmol/l) $\times$ 351.44 g/mol (Am80 molecular weight).

## Analyses of cell proliferation, granulocytic morphologic differentiation, and neutrophil phagocytosis

As described previously (Luo *et al*, 2007; Lou *et al*, 2013), cell proliferation was determined by cell counting, whereas Wright-Giemsa stain was used to assess granulocytic morphologic differentiation as well as neutrophil phagocytosis in parallel with Gram stain (see Gram stain analysis).

## Luminol chemiluminescence analysis of ROS production

Reactive oxygen species (ROS) production of neutrophils was measured by using cellular luminol chemiluminescence (LCL) assay (Wei *et al*, 2010) on a luminometer (GloMax®-Multi Microplate Multimode Reader, Promega, Madison, WI, USA). Two most commonly used ROS stimulators mimicking bacterial infection of neutrophils, fMLP (*N*-formyl-methionyl-leucylphenylalanine) and PMA (phorbol myristate acetate) (Braga *et al*, 2006; Barnes *et al*, 2012), were used to induce ROS production. Except primary human PB neutrophils that were directly used for analyzing ROS production, every $5 \times 10^5$ cells were cultured for 3–12 days with corresponding medium (0.2 ml) at least in triplicate in white plastic, non-coated 96-well plate (Falcon, Monroe, NC). Immediately before LCL assay, cells were treated with 0.5 mM luminol (5-amino-2,3-dihydro-1,4-phthalazinedione, Sigma-Aldrich) in the presence of 4 U/ml horseradish peroxidase (HRP; Sigma-Aldrich) at 37°C for 30 min, as described (Zhu *et al*, 2005). Then, fMLP or PMA was dispensed at a final concentration of 1 μM fMLP (Sigma-Aldrich) (Barnes *et al*, 2012) or 0.5 μg/ml PMA (Sigma-Aldrich) (Ding *et al*, 2013) by the injector built in the plate reader. Baseline value of samples without stimulator was assessed

three times immediately before the initiation of reactions in the presence of stimulators. Based on the transient response nature of LCL assay, chemiluminescence was measured sequentially and recorded at 60-s intervals for 15 min for fMLP plate while 90 min for PMA plate. The relative light units (RLU) were assessed with at least triplicate wells, and the area under the curve (AUC) was used to compare the chemiluminescence changes in control samples.

## Flow cytometric analysis

Mouse anti-human CD66-PE (Catalog #: 551480; recognizing CD66a, CD66c, CD66d, and CD66e subunits), PE IgG (Catalog #: 555787), mouse anti-human CD18-FITC (Catalog #: 557156), and FITC IgG (Catalog #: 555786) were from BD Biosciences. Following manufacturer's recommendations, the dilutions of antibodies and control IgG for experiments were 1:5. Flow cytometric analysis was performed as previously described (Ding *et al*, 2013). Data were acquired and analyzed with FlowJo software (version 7.6.5; Tree star, Ashland, OR).

## Antibody neutralization of neutrophil ROS production and bacterial killing

Mouse anti-human CD18 (Catalog #: CBL158; Millipore, Billerica, MA) and CD11b (Catalog #: MAB1387Z, Millipore) antibodies were used in antibody neutralizing analysis. Normal IgG1-κ (Catalog #: 553447, BD) was used as additional control. For antibody neutralization of ROS production, fresh primary PB neutrophils ($2 \times 10^5$) collected from normal human donors were first incubated with 1 μg/ml (1:1,000 dilution) anti-CD18, or anti-CD11b, or combined anti-CD18 and anti-CD11b antibodies at RT for 20 min, respectively. In parallel, neutrophils were incubated with 1 μg/ml (1:1,000 dilution) normal mouse IgG as controls. The resulting reaction mixtures were then subjected to the procedures of ROS production assay. For antibody neutralization of bacterial killing, primary PB mononuclear cells collected from normal human donors were cultured with MM-HSCCM medium for 3 days to induce granulocytic differentiation. Cells were then incubated with 1 μg/ml (1:1,000 dilution) anti-CD18 antibody for each $2 \times 10^5$ cells or 1 μg/ml (1:1,000 dilution) normal mouse IgG for each $2 \times 10^5$ cells at RT for 20 min. The resulting reaction mixtures were then immediately processed for bacterial killing assay.

## Quantitative real-time polymerase chain reaction (qRT–PCR)

mRNA was extracted from cells with RNeasy Mini Kit (Qiagen, Valencia, CA), and converted into cDNA with RevertAid First Strand cDNA Synthesis kit (Thermo Scientific, Pittsburgh, PA). qRT–PCR analysis, as described before (Lou *et al*, 2013), was performed in 384-well optical plates on the 7900HT FastqRT–PCR System (Applied Biosystems, Carlsbad, CA). Amplification condition was set up according to the manufacturer's instructions (Invitrogen, Grand Island, NY). qRT–PCR was performed with an initial denaturation at 95°C for 15 min, followed by 40 consecutive thermal cycles (94°C, 15 s; 59°C, 30 s; and 72°C, 30 s) and a final dissociation curve cycle (95°C, 15 s; 60°C, 15 s; and 95°C, 15 s). Each of the samples was assessed at least in triplicate, and the fluorescence

intensities were normalized with ROX dye as reference standard. Relative mRNA abundance was calculated with the $\Delta\Delta C_T$ method, as described in manufacturer's manual (Applied Biosystems), by normalizing observed Ct values to those of GAPDH and assuming amplification efficiencies of 2. Fold change was then determined by comparing the value of experimental group with that of control group; the values for control samples were represented as 1 in figures, as described (Hansen *et al*, 2014).

### Calculating equivalent doses of GCSF and Am80 between human and mouse for *in vivo* studies

Following dose conversion guidelines for human equivalent doses (HED) set by the FDA (www.fda.gov/downloads/drugs/guidance complianceregulatoryinformation/guidances/ucm078932.pdf), HED of GCSF and Am80 were calculated, respectively (see Fig 4B). Subcutaneous injection of GCSF to neutropenia patients is 5–10 µg/kg/day for 7–14 days (Amgen, www.neupogen.com). The calculation equation is (Reagan-Shaw *et al*, 2008):

$$HED\ (mg/kg) = animal\ dose\ (mg/kg) \times K_m\ (mouse) \div K_m\ (human)$$

whereas $K_m$-mouse = 3 and $K_m$-human = 37. This method was also used to estimate HED of Am80. Oral doses of Am80 for APL patients are 6–9 mg/m$^2$/day for maximum 58 days (Tobita *et al*, 1997).

### Preparation of GCSF and Am80 for *in vivo* studies

GCSF was diluted with sterile 0.1% BSA-PBS for preparing of 50 µg/µl stock solution and stored at −80°C. Immediately before subcutaneous (sc) injection of mice, the GCSF stock solution was diluted with 0.1% BSA-PBS to low and medium doses of working solutions, that is, 25 and 50 µg/kg, equivalent to human GCSF doses of 2 and 4 µg/kg, respectively. Am80 was dissolved in 100% ethanol to generate 5 mg/ml stock solution and stored at −80°C. Immediately before intragastric (ig) administration to mice, the Am80 stock solution was first mixed with one volume of corn oil, followed by a dilution with eight volumes of sterile water in order to make low and medium doses of working solution.

### Mice, mouse CCIN models, and mouse work

Animal studies were performed according to the guidelines of protocols approved by CHLA Institutional Animal Care and Use Committee. Normal C57BL/6J mice (female, 16–22 g, 6–8 weeks old) were purchased from Jackson Laboratory (Bar Harbor, ME) and randomly divided into different groups for tests. Mice were housed in CHLA institutional animal facility, which had an air-conditioned environment at 25°C together with a shifting light–dark schedule of each 12 h. Mice were provided with access to food and water freely. To induce mouse CCIN, as described (Ding *et al*, 2013), experimental mice received single 200 mg/kg intraperitoneal dose of cyclophosphamide (CPA, Baxter Healthcare Co. Deerfield, MA) at day 0, while injecting control mice with PBS. Six different models, covering the neutrophil decrease, neutrophil recovery, and a full cycle of mouse CCIN, were used in this study. Regimen treatments for different mouse CCIN models were applied once a day, with Am80 by ig or GCSF by sc or vehicle by ig. To monitor the levels of total

leukocytes and neutrophils before or during the experiments, ~50 µl PB was collected from the tail vein of live mice and analyzed with Vetscan HM5 (Abaxis, Union City, CA) by following the manufacture's protocol. Based on different dose-schedule-infection models, mice were challenged with bacterial CFU of $6 \times 10^6$ (neutrophil decrease model), or $9 \times 10^6$ (neutrophil recovery model), or $5 \times 10^6$ (survival model covering a full cycle of mouse CCIN), with intravenous injection via tail vein. The different numbers of bacterial CFU used for infecting different mouse CCIN models were determined, respectively, by assessing the lethal amounts of bacteria that induced death of vehicle mice ~24 h post-infection. The first bacterial infection was for 3 or 16 h or up to 13 days, depending on different models. A possible second infection after 6 days of the first infection was only performed in survival models, by which mice surviving on day 9 were injected with bacteria for 15 min to assess bacterial clearance, neutrophil phagocytosis, and neutrophil maturation. Body weight of mice was recorded every 2 days.

### Survival evaluation of CCIN mice

Survival of CCIN mice was tested throughout a full cycle of neutropenia undergoing perpetual systemic intravenous bacterial infection. Occurrence of moribund or dead mice was monitored three times daily. Mice were considered moribund when at least two of following clinical signs were observed: impaired ambulation, inability to remain upright, decreased or labored breathing, or no response to external stimuli, as described (Gresham *et al*, 2000).

### Assessment of neutrophil recovery in PB and BM of CCIN mice

Mouse PB (~1.5 ml) was collected by cardiac puncture immediately after mouse euthanasia. Neutrophils were purified by use of proportionally reduced volume of Ficoll–Paque Premium 1.084, as described before (Ding *et al*, 2013). The purity of collected PB neutrophils was > 95%, as measured by morphologic analysis with Giemsa stain. Mouse BM cells were collected through flushing femurs with 28-gauge syringe filled with PBS. BM cells were then purified with gradient Percoll (GE Healthcare, Piscataway, NJ). To adequately separate mouse BM cells into different layers containing of more versus less mature neutrophils, Percoll gradient solution was prepared to consist of three densities of 50, 60 and 71%. Mouse BM cells loaded on this gradient solution were centrifuged at $400 \times g$ for 40 min at RT. Four layers of mouse BM cells generated by gradient separation were subjected to quantitative and granulocytic morphologic analyses. Recovered neutrophils were calculated as follows: neutrophil number = total cell number × the percentage of granulocytes in total cells.

### Bacteria, bacteria preparation, and monitoring of bacterial growth

Gram-positive *S. aureus* bacteria (ATCC-27217; ATCC, Manassas, VA) causing bacteremia in neutropenic patients with cancer (Gonzalez-Barca *et al*, 2001) were used to infect mice, as described previously (Ding *et al*, 2013). *S. aureus* were stored at −80°C in broth containing 50% (v/v) glycerol and renewed every 3 months. For infection, log-phase growing bacteria (0.3–0.5 at OD$_{600}$) were

centrifuged at $1,300 \times g$ for 10 min at 4°C and washed twice with sterile saline. The bacterial pellet was re-suspended thoroughly with appropriate volume of saline before injection. To monitor the number of bacteria injected into the mice, an aliquot of bacterial suspension used for infecting mice was diluted in a series of 10, 100, 1,000, and 10,000 times (20 μl each), and then plated onto nine blood agar plates for each of dilutions, respectively. The numbers of injected bacteria were estimated by counting of colonies after incubation of bacteria overnight, as described (Ding et al, 2013).

### Bacterial killing

In vitro bacterial killing was performed as described before (Ding et al, 2013). To assess in vivo bacterial killing, PB was collected at different time points from either bacteria-infected live mice by phlebotomy of tail vein, or through cardiac puncture with 27½-gauge syringe immediately after euthanasia of bacteria-infected mice. The samples were centrifuged at $100 \times g$ for 5 min at 4°C to recover the supernatants containing extracellular bacteria. The equal amounts of supernatant were then enumerated by plating a series of 10, 100, 1,000, and 10,000 dilutions (20 μl each) onto 3–9 blood agar plates for each of dilutions, respectively. The agar plates were incubated at 37°C for 14 h, and the numbers of viable bacteria at each time point were determined by counting of colonies, as described (Ding et al, 2013). Spleens or hearts were collected immediately after euthanasia of mice, and rinsed briefly with sterile saline. These organs were then homogenized in the presence of sterile saline (0.05 g/1 ml), respectively. The supernatant was recovered by centrifuging at $150 \times g$ for 5 min at 4°C. The amounts of bacteria infecting organs were enumerated by plating a series of 1, 10, 100, and 1,000 dilutions of supernatant (20 μl each) onto three blood agar plates for each of dilutions, respectively. The agar plates were incubated at 37°C for 14 h, and the numbers of viable bacteria in spleens and hearts were determined by counting of colonies.

### Gram stain of neutrophil phagocytosis and bacterial clearance

Mice surviving on day 9 were divided into two groups: one challenged by bacterial infection for 15 min before euthanasia, and other one injected with vehicle solution. PB was collected by cardiac puncture, and neutrophils were purified by Ficoll separation without further hydrolysis of red cells for maximally preserving the morphology of neutrophils and infected bacteria. Each 20 μl PB neutrophils was cyto-centrifuged onto micro-slides, and fixed with methanol for 5 min. The Gram stain was performed at CHLA Pathology Research Core, and the slides were examined under the oil immersion lens of Zeiss Axioplan microscope.

### Transmission electron microscopy analysis of mouse PB neutrophils

PB was collected from mice surviving on day 11 by cardiac puncture immediately after euthanasia. PB neutrophils were purified by Ficoll isolation without further hydrolysis of red cells for preserving neutrophil structures. Electron microscopy analysis, as described (Ding et al, 2013), was performed at CHLA Pathology Research

## The paper explained

### Problem

Granulocyte colony-stimulating factor (GCSF) has been the primary drug used for treatment of cancer chemotherapy-induced neutropenia (CCIN) for over two decades. Although GCSF can profoundly increase the number of neutrophils to reduce the duration of neutropenia, these neutrophils are inadequately differentiated with impaired microbicidal function. Thus, CCIN remains challenging with substantial morbidity and mortality, as well as healthcare cost. Moreover, there is a potential association between the clinical use of GCSF and the development of myeloid malignancy. As such, an alternative mode of treatment is needed for CCIN, which can induce sufficient numbers of functional neutrophils while preventing possible risk of myeloid malignancy.

### Results

Am80 is a retinoic acid (RA) agonist. It enhances granulocytic differentiation by selectively activating transcription factor RA receptor alpha (RARα) to modulate RA-target gene expression. We discovered that Am80-GCSF combination altered transcription of RA-target genes coordinating proliferation with differentiation to develop neutrophil innate immunity. This led to generation of functional neutrophils capable of fighting infection in both normal and malignant primary human hematopoietic specimens. Specific antibodies against neutrophil innate immunity abolished neutrophil bactericidal activities induced by Am80-GCSF. Such Am80-GCSF synergy was further evaluated using six different mouse CCIN models with varying dose, treatment schedule, and duration of intravenous bacterial infection. The results demonstrated that without causing myeloid overexpansion, Am80-GCSF coordinated myeloid expansion with differentiation to generate sufficient numbers of functional neutrophils that significantly reduced infection and infection-related mortality in CCIN mice.

### Impact

This work reveals a novel Am80-GCSF synergy in generating functional neutrophils to reduce CCIN-associated infection and mortality, thus proposing an advanced therapy that has potential efficacy for CCIN treatment.

Core. In brief, PB neutrophils were first fixed with 2% glutaraldehyde in 0.1 M phosphate buffer, and then post-fixed with 1% osmium tetroxide ($OsO_4$) in 0.1 M phosphate buffer for 1 h, following dehydration processes with series of alcohol. After embedded with epoxy resin in beam capsules and polymerized at 60°C oven for 72 h, the samples were cut into ultrathin section and stained with uranyl acetate and lead. Ultrastructural image of neutrophils were performed on a Morgagni 268 transmission electron microscope.

### Statistical analysis

Descriptive statistics, including means, standard deviations, were computed at each time point for each experimental and control group, using Student's unpaired two-tailed t-test. P-values of 0.05 or less were considered statistically significant. Statistical data shown are means ± standard deviations (SD) representing at least triplicate assays. The survival of CCIN mice was statistically calculated with Kaplan–Meier survival plots analysis (Becks et al, 2010; Lerche et al, 2011), whereas the differences among the groups were compared by using a log-rank test (Lerche et al, 2011). Differences

in survival times were considered significant if the *P*-value was < 0.05.

**Expanded View** for this article is available online.

## Acknowledgements

This work was supported by grants to L. Wu, including the National Institutes of Health (R01 CA120512, R01 ARRA-CA120512) and Saban Inspire Innovation Award, CHLA. We thank Dr. Yao-Te Hsieh's assistance in culturing primary leukemoblasts; Dr. Jennifer Dien Bard for helping in the Gram stain of *S. aureus*; Drs. Jian Jian Li, Xiaokun Zhang, and Martin Broome for the critical review of this manuscript.

## Author contributions

LL and XQ performed design, data collection, analysis, and interpretation and wrote the manuscript. WS collected clinical samples and performed data interpretation. HA-A collected clinical samples and involved in data interpretation and manuscript preparation. SL and HZ performed data collection, analysis, and interpretation. NVP provided experimental materials, performed data analysis and interpretation, and involved in manuscript preparation. AZ performed data collection and involved in manuscript preparation. HS performed data analysis and involved in manuscript preparation. KS provided experimental materials, performed data interpretation, and involved in manuscript preparation. Y-MK provided technical support and involved in manuscript preparation. SK collected clinical samples and performed data interpretation. QH and DW performed design and involved in manuscript preparation. LW contributed to conception and design, data analysis and interpretation, manuscript writing, financial support, and final approval of manuscript.

## Conflict of interest

The authors declare that they have no conflict of interest.

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
