## [Review Process File · EMBO Molecular Medicine]

Am80-GCSF synergizes myeloid expansion and differentiation to generate functional neutrophils that reduce neutropenia-associated infection and mortality

Lin Li, Xiaotian Qi, Weili Sun, Hisham Abdel-Azim, Siyue Lou, Hong Zhu, Nemani V. Prasadarao, Alice Zhou, Hiroyuki Shimada, Koichi Shudo, Yong-Mi Kim, Sajad Khazal, Qiaojun He, David Warburton, and Lingtao Wu

Corresponding author: Lingtao Wu, CHLA/USC

Review timeline:

Submission date:	22 March 2016
Editorial Decision:	19 May 2016
Revision received:	19 August 2016
Editorial Decision:	01 September 2016
Revision received:	18 September 2016
Accepted:	19 September 2016

Transaction Report:

Editor: Céline Carret

1st Editorial Decision

19 May 2016

Thank you for the submission of your manuscript to EMBO Molecular Medicine. We have now heard back from the three referees whom we asked to evaluate your manuscript. As you will see from the reports below, the referees find the topic of your study relevant and of interest. However, they raise substantial concerns on your work, which should be convincingly addressed in a major revision of the present manuscript.

You will see that referees 1 and 2 are particularly concerned about the technical aspect of the work and would like to see the data strengthened, especially regarding the first half of the manuscript. Referees 1 and 3 highlight the relevance of the second part of the study (the *in vivo* data), but referee 3 suggests increasing the clinical aspect of the work by providing a titration experiment, which we agree would improve the study.

Given that all of them find the message novel and interesting we would be willing to consider a revised manuscript with the understanding that the referee concerns must be fully addressed and that acceptance of the manuscript would entail a second round of review.

I should remind you that it is EMBO Molecular Medicine policy to allow a single round of major revision only and that, therefore, acceptance or rejection of the manuscript will depend on the completeness of your responses included in the next, final version of the manuscript. I realize that

addressing the referee comments in full would involve a lot of additional experimental (and grammatical/spelling) work and I am uncertain whether you will be able (or willing) to return a revised manuscript within the 3 months deadline and I would also understand your decision if you chose to rather seek rapid publication elsewhere at this stage.

Should you decide to revise your article for EMBO Molecular Medicine, revised manuscripts should be submitted within three months of a request for revision; they will otherwise be treated as new submissions, except under exceptional circumstances in which a short extension is obtained from the editor. Please make sure to format your article according to our guidelines and provide all the requested editorial amendments as listed below.

I look forward to seeing a revised form of your manuscript as soon as possible.

Should you find that the requested revisions are not feasible within the constraints outlined here and choose, therefore, to submit your paper elsewhere, we would welcome a message to this effect.

***** Reviewer's comments *****

Referee #1 (Comments on Novelty/Model System):

See my comments to the authors below

Referee #1 (Remarks):

Neutrophils are key players in infection defense. G-CSF is therefore used since >20 years to shorten the period of neutropenia after chemotherapy. The authors demonstrate the surprising finding that G-CSF alone is able to induce the development of neutrophils from precursors in vitro and in experimental animals in vivo, yet these cells are functionally immature by a number of measures and thus are unable to protect mice from experimental bacterial infections. The authors then show, that stimulating the retinoic acid system with the artificial trigger Am80 is able to induce neutrophil maturation alone, yet in insufficient numbers. However, when provided in combination with G-CSF, Am80 is able to produce sufficiently high numbers of fully mature neutrophils in vivo, that are protective.

While the main finding is very interesting, the manuscript suffers from several technical flaws, data that are inconsistent with the provided interpretation or not convincing and, not the least, very complicated writing with partly poor English. Overall this serves to significantly dampen the enthusiasm of this reviewer.

Key issues are listed below:

Major:

- 1) Generally the image quality is very poor (e.g. Fig. 1A ii, 2D v, 4D/E, 6A/B and supplementals). This does not allow to distinguish morphological differences in the nuclear shapes of cells, which are used by the authors to highlight inefficient maturation of the cells.
- 2) In Fig. 1B the collection of measured gene regulations is not clear and seems to vary at will (e.g. CD66c/b on day 1, CD66c and CD11b on day 2 and CD66c and CD18 on day 6). There is no rationale provided, for why this was chosen. Generally, the differences in gene regulation are also often very small (e.g. CD18 (day 6): Am80 ~0.7, Am80+GCSF ~ 1.2. Highly significant by statistics, yet very small in effect).
- 3) The interpretation of the data in Fig. 1B is not supported by the figure. Example 1: "We found that compared to Am80 in the early and late differentiation induction stages, Am80-GCSF induced significantly higher expression of RA-target genes that regulated growth inhibition and granulocytic differentiation (Fig. 1B; day 1 vs. day 2 vs. day 6), including RAR 2 (Alvarez et al, 2007; Soprano et al, 2004), C/EBP (Lekstrom-Himes, 2001; Park et al, 1999), CD11b (Park et al, 1999), CD66 (Park et al, 1999), and CD18 (Bush et al, 2003)." Data show: Am80 values higher in RAR 2, C/EBP, CD66c and CD11b (day 2). Example 2: "GCSF only induced higher expression of RAR 2 and C/EBP at day 6 than did Am80". Data show, that also CD66c and CD18 are higher.
- 4) Fig. 1C: Why was "CD66" measured and not a more specific version such as CD66b, that was also measured in the qPCR data?

5) Why were normal neutrophils from peripheral blood tested in Fig. 1D? This does not fit to any of the other data of the MS, which work with cell lines or CD34+ precursors.

6) Fig. 3E: the interpretation "among CCIN mice groups, both low and medium doses of Am80 sustained significantly higher numbers of neutrophils than did G-CSF in PB" is not supported by the data. Data on low dose show: G-CSF: $22 \times 10^4/\text{ml}$, Am80: $22 \times 10^4/\text{ml}$. This is identical. Also the notion "Am80-G-CSF were similar to Am80 in PB but comparable to G-CSF in BM" is not at all supported by the shown values.

7) Fig. 6D: "Compared to G-CSF mice, neutrophil production in Am80-G-CSF mice was not associated with significant loss of body weight". The data show a loss from ~20g to below 15 g. This is >25%, so very significant. Typically, a loss of >20% requires to terminate the experiment. Also in the same experiment: why is there weight loss at all in the control experiment, when there is neither CPA treatment nor infection?

Minor:

1) Fig. 2E, day 6, CD11b: X-Axis is strange

2) Fig. 3J, 16h PB: Y-axis labelling missing

3) Fig. 4C, 3hr: differences between G-CSF and A+G very small, even if measured significant.

Referee #2 (Comments on Novelty/Model System):

This is probably the best available model, short of a clinical trial in humans.

Referee #2 (Remarks):

The authors propose that the combination of a retinoic acid agonist, Am80, and recombinant human G-CSF promote a more functional proliferative and differentiation program, which may be useful in chemotherapy-induced neutropenia. This is a data-rich manuscript.

Shortcomings center on the work is done mostly in mouse, not humans, and the investigators do not consider that there may be nuances in RA/G-CSF signaling between species.

While pharmacologic dissection of granulopoiesis is shown to be informative, it does not reach the same level of rigor as RNAi silencing or gene targeting.

The authors refer to G-CSF and its relationship to myeloid malignancies - the precise contribution to leukemia in severe congenital neutropenia, severe aplastic anemia, and breast cancer are either controversial or specific to the underlying disease and need for chronic administration of G-CSF. It will be very difficult to demonstrate that the addition of Am80 would mitigate this risk, as the mouse studies fail to model this phenomenon.

Figure 7 should include G-CSF as a separate stimulus for granulopoiesis.

Referee #3 (Remarks):

The paper by Li et al "Am8-G-CSF synergizes..." is of great interest, as it suggests a solution to an important clinical problem, neutropenia associated infection following chemotherapy.

It uses a combination of human and murine studies, but the most compelling data, protection and survival from lethal infection is in the mouse.

The key data in the paper is Figure 5, which documents the marked survival advantage of the combination compared to either drug alone. This is impressive at the doses used. However the doses used are difficult to extrapolate across species, as affinity of receptor, metabolism and other

variables necessarily differ.

So a key experiment, which would improve the paper, and enhance its clinical relevance, is to titrate the doses of AM80 and GCSF. This is relevant as it would help define a 'therapeutic window'. Compliance of all drugs is variable and GCSF causes a lot of bone pain, hence compliance is variable, and so dosing is also.

The lowest doses of AM80 and GCSF which synergize would be of significant clinical relevance.

1st Revision - authors' response

19 August 2016

Reviewer 1's comments and our responses

1. "Generally the image quality is very poor (e.g. Fig. 1A ii, 2D v, 4D/E, 6A/B and supplementals)."

Response

We apologize for those low-quality images. We have now provided high-resolution images that show a clear contrast between nucleus and cytoplasm, which allow distinguishing the degrees of neutrophil morphological differentiation between more mature vs. less mature neutrophils.

2. "In Fig. 1B the collection of measured gene regulations is not clear and seems to vary at will (e.g. CD66c/b on day 1, CD66c and CD11b on day 2 and CD66c and CD18 on day 6). There is no rationale provided, for why this was chosen."

Response

We regret for not providing sufficient information and clear interpretation before. Because Am80 promoted granulocytic differentiation by selectively activating RAR α to alter transcription of RA-target genes, we investigated RAR α -dependent gene expression modulated by Am80-GCSF in generating functional neutrophils. We found that 6 of 12 different RAR α target genes that play key roles in granulocytic differentiation were dynamically modulated by Am80-GCSF at different differentiation induction stages, including tumour suppressor *RAR β ₂*, terminal granulocytic differentiation regulator *C/EBP ϵ* , as well as neutrophil innate immunity regulators *CD66c*, *CD66b*, *CD11b*, and *CD18*. Interestingly, whereas both *RAR β ₂* and *C/EBP ϵ* were consistently induced in all differentiation induction stages, they were associated with different transcriptional inductions of innate immunity regulators, i.e., *CD66c* throughout all time, *CD66b* in the early stage, *CD11b* in the middle stage, and *CD18* in the late stage, showing that Am80-GCSF may mediate a course of neutrophil differentiation-associated innate immunity development. In the revised manuscript, we have now provided rationale and interpretation about such transcriptional inductions of those innate immunity regulators at different differentiation induction stages (Results: page 5, lines 12 to page 6, lines 1 to 3).

3. "The interpretation of the data in Fig. 1B is not supported by the figure. Example 1: "We found that compared to Am80 in the early and late differentiation induction stages, Am80-GCSF induced significantly higher expression of RA-target genes that regulated growth inhibition and granulocytic differentiation (Fig. 1B; day 1 vs. day 2 vs. day 6), including RAR β ₂ (Alvarez et al, 2007; Soprano et al, 2004), C/EBP ϵ (Lekstrom-Himes, 2001; Park et al, 1999), CD11b (Park et al, 1999), CD66 (Park et al, 1999), and CD18 (Bush et al, 2003)." Data show: Am80 values higher in RAR β ₂, C/EBP ϵ , CD66c and CD11b (day 2). Example 2: "GCSF only induced higher expression of RAR β ₂ and C/EBP ϵ at day 6 than did Am80". Data show, that also CD66c and CD18 are higher."....."Generally, the differences in gene

regulation are also often very small (e.g. CD18 (day 6): Am80 ~0.7, Am80+GCSF ~ 1.2. Highly significant by statistics, yet very small in effect)."

Response

We agree with the reviewer that our data showed that: a) Am80-GCSF promoted significantly higher expression of target genes than did Am80 in the early and late differentiation induction stages, while Am80 inducing higher expressions in the middle stage; and b) although both GCSF and Am80-GCSF are highly significant by statistics in promoting transcriptions of *RARβ₂*, *C/EBPε*, *CD66c*, and *CD18* than did Am80 in the late differentiation induction stage, such effects were relatively small. We apologize for any confusion that may have introduced in the previous version and have now clarified data presentation in the revised manuscript (Results: page 6, lines 3 to 9).

4. "Fig. 1C: Why was "CD66" measured and not a more specific version such as CD66b, that was also measured in the qPCR data?"

Response

We regret for not providing necessary rationale before. It is known that either CD66a, CD66b, CD66c, or CD66d can independently transmit signals in neutrophils, whereas co-expression of different CD66 subunits with CD18 surface markers are associated with the critical development of CR3-dependent neutrophil innate immunity.¹⁻³ Since Am80-GCSF induced significantly higher transcriptions of *CD66c*, *CD66b*, and *CD18* than did Am80 in both early and late differentiation induction stages, we chose to examine the corresponding protein levels of CD66-CD18 modulated by Am80-GCSF. We have now highlighted this point in the revised manuscript (Results; page 6, lines 9 to 14).

5. "Why were normal neutrophils from peripheral blood tested in Fig. 1D? This does not fit to any of the other data of the MS, which work with cell lines or CD34+ precursors."

Response

In the previous version, data in both Fig. 1D and 1E were derived from normal peripheral blood (PB) specimens. We regret for not clearly emphasizing the use of these normal PB neutrophils before. Whereas NB4 cell line was used in a few parallel tests (Appendix-1 Figs. S1A, B; S2), normal primary human hematopoietic specimens were used in the studies (Fig. 1, Appendix-1 Fig. S1C, D), including both normal primary human hematopoietic CD34+ precursors derived from umbilical cord blood and normal PB specimens collected from normal human donors. By using these normal specimens, we determined that neutrophils induced by Am80-GCSF from normal primary PB mononuclear cells gained CR3-dependent innate immunity, similar to those in normal primary human PB neutrophils or neutrophils induced from CD34+ cells. We have now highlighted the use of these normal specimens in the revised manuscript. Moreover, we have provided new data derived from normal PB specimens, designated as new Fig. 1D. Together, the newly arranged Fig. 1D-F data showed that neutrophils induced by Am80-GCSF from normal primary human PB mononuclear cells display effective innate immunity, mimicking bactericidal activities observed in normal primary human PB neutrophils. All of these changes have now been included in the revised manuscript (Results: page 6, lines 12 to 1 from the bottom).

6. "Fig. 3E: the interpretation "among CCIN mice groups, both low and medium doses of Am80 sustained significantly higher numbers of neutrophils than did GCSF in PB" is not supported by the data. Data on low dose show: G-CSF: 22×10^4 /ml, Am80: 22×10^4 /ml. This is identical. Also the notion "Am80-GCSF were similar to Am80 in PB but comparable to GCSF in BM" is not at all supported by the shown values."

Response

We apologize for any confusion that may have introduced in the previous version. The interpretations for Fig. 3E data were focusing on the “numbers of neutrophils” (see “neutrophils” section) but not the total numbers of cells (see “Total cells” section). We have now clarified data presentation in the revised manuscript where this original Fig. 3 has been arranged as Fig. 4 because of an added new Fig. 3 (Results: page 10, lines 10 to 15).

Moreover, in order to highlight our focus on recovering “numbers of neutrophils,” we have now deleted the statistical significance markers that were previously presented in original Fig. 3E and 3I under the section of “neutrophils, (%)”.

7. “Fig. 6D: “Compared to GCSF mice, neutrophil production in Am80-GCSF mice was not associated with significant loss of body weight”. The data show a loss from ~20g to below 15 g. This is >25%, so very significant. Typically, a loss of >20% requires to terminate the experiment. Also in the same experiment: why is there weight loss at all in the control experiment, when there is neither CPA treatment nor infection?”

Response

- A) We agree with the reviewer that a loss of >20% body weight could be one of moribund signs for euthanizing a mouse. However, we found that significant loss of body weight by bacterial infection in GCSF mice was not always immediately associated with other clinical moribund signs. Therefore, to evaluate infection-induced mortality in this study, mice were considered moribund when at least two of following clinical signs were observed: impaired ambulation, inability to remain upright, decreased or labored breathing, or no response to external stimuli, as described before.⁴ We have now highlighted this standard in the revised manuscript (Materials and Methods: page 24, 2nd paragraph, lines 3 to 6).
- B) In this study, control mice were not injected with CPA but were subjected to bacterial infection in parallel to all other test groups’ mice. On the other hand, all mice in four different test groups, including vehicle, GCSF, Am80, and Am80-GCSF, were subjected to both CPA injection and bacterial infection. Thus, control mice did encounter loss of body weight under perpetual systemic intravenous bacterial infection. The design for control and test groups in different mouse models has now been provided in Appendix-1 Table S3 as well as stated in the related figure legends. Furthermore, the design for with or without CPA injection of mice has also been emphasized in the revised manuscript (Results: page 9, 2nd paragraph, line 1 from the bottom to page 10, lines 1 to 2).

8. “Fig. 2E, day 6, CD11b: X-Axis is strange.”

Response

We apologize for this error and have now made correction in the figure. Because we have included new drug titration data that are designated as new Fig. 3, this original Fig. 2E has now been rearranged as Fig. 2B in the revised manuscript.

9. “Fig. 3J, 16h PB: Y-axis labelling missing.”

Response

Y-axis of 3 hr (section i) and 16 hr (section ii) in Fig. 3J share the same legend “bacteria (10³/ml)”. To avoid confusion, we have now added a legend to Y-axis of 16 hr (section ii). Because we have

now provided new drug titration data that are designated as new Fig. 3, this original Fig. 3J has now been arranged as Fig. 4J in the revised manuscript.

10. “Fig. 4C, 3hr: differences between G-CSF and A+G very small, even if measured significant.”

Response

We agree with the reviewer’s comments. Although bacterial killing induced by Am80-GCSF after 3 hr of bacterial infection was significant compared to GCSF mice ($P < 0.01$), the numbers of killed bacteria by Am80-GCSF mice were relatively small. However, after 16 hr of infection, the difference had increased markedly ($P < 0.001$). This result suggests that neutrophils induced by Am80-GCSF are capable of continuously killing bacteria in a longer period of transient infection. This original Fig. 4C has now been arranged as Fig. 5C due to an added new Fig. 3.

11. “very complicated writing with partly poor English.”

Response

We apologize for those language problems. We have now made corrections in the revised manuscript by using simpler and shorter sentences with close attention to syntax.

Reviewer 2’s comments and our responses

1. “Shortcomings center on the work is done mostly in mouse, not humans, and the investigators do not consider that there may be nuances in RA/GCSF signaling between species.”

Response

We thank the reviewer for pointing this out. To enhance clinical relevance of this study by providing a 'therapeutic window' reference for the future clinical study of Am80-GCSF treatment of cancer chemotherapy-induced neutropenia (CCIN), we have now titrated the dose ranges of Am80 when combined with GCSF in primary acute myeloid leukemia (AML) patient specimens. We have now defined that several dose combinations of Am80-GCSF effectively induce functional neutrophils while suppressing leukemic growth, likely through mediating an altered transcription of RA signaling molecules in AML patient specimens. We have now presented these new data in a new Fig. 3 in the revised manuscript (see details in the Response to reviewer 3’s comments).

2. “While pharmacologic dissection of granulopoiesis is shown to be informative, it does not reach the same level of rigor as RNAi silencing or gene targeting.”

Response

One of the findings derived from this study has revealed a differential course of proliferation vs. differentiation in primary human specimens, as shown by: a) in normal primary human hematopoietic precursors, Am80-GCSF synergizes active proliferation with effective granulocytic differentiation to generate significantly larger amount of functional neutrophils than does Am80; and b) Am80-GCSF produces functional neutrophils while inhibiting malignant growth in primary human AML specimens. These results raise a question: How can Am80-GCSF modulate such differential processes in normal vs. malignant cells to coordinate innate immunity development with neutrophil production? We agree with the reviewer that in order to determine the mechanisms of such differentially synergized regulatory processes by Am80-GCSF, a future rigor study is needed to define an array of transcription factors that coordinate with RAR α at distinct developmental

stages, through RNAi silencing and/or gene targeting in normal and malignant cells in the presence or absence of Am80-GCSF, respectively. We have now thoroughly discussed this needed future study in the revised manuscript (Discussion: page 16, 2nd paragraph, lines 1 to 3 to page 18, lines 1 to 2).

3. “The authors refer to GCSF and its relationship to myeloid malignancies - the precise contribution to leukemia in severe congenital neutropenia, severe aplastic anemia, and breast cancer are either controversial or specific to the underlying disease and need for chronic administration of GCSF. It will be very difficult to demonstrate that the addition of Am80 would mitigate this risk, as the mouse studies fail to model this phenomenon.”

Response

We agree with the reviewer that our mouse models have the limitations, which mainly mimic CCIN-associated infection and mortality rather than GCSF-induced possible myeloid overexpansion. However, compared to GCSF *in vivo*, Am80-GCSF induces sufficient numbers of functional neutrophils while preventing myeloid overexpansion (Figs. 5-7). Also, Am80-GCSF can induce functional neutrophils while inhibiting leukemic growth in cultured AML specimens (Fig. 3). Therefore, further clinical studies of Am80-GCSF for CCIN treatment may be critical to evaluate directly whether Am80-GCSF combination mitigates the risk of myeloid malignance in human.

4. “Figure 7 should include G-CSF as a separate stimulus for granulopoiesis.”

Response

We thank the reviewer for pointing this out and have now included a GCSF section related to transcription of RA target genes in the figure. Due to the addition of a new Fig. 3, the original Fig. 7 has now been designated as Fig. 8 in the revised manuscript.

Reviewer 3’s Comments and our responses

“The key data in the paper is Figure 5, which documents the marked survival advantage of the combination compared to either drug alone. This is impressive at the doses used. However the doses used are difficult to extrapolate across species, as affinity of receptor, metabolism and other variables necessarily differ. So a key experiment, which would improve the paper, and enhance its clinical relevance, is to titrate the doses of AM80 and GCSF. This is relevant as it would help define a 'therapeutic window'.”

Response

We thank the reviewer for calling our attention to this important issue. Clinical GCSF doses have been recognized worldwide in the past over 2 decades. Also, the medium human plasma concentration-dose of GCSF (25 ng/ml) in mediating granulocytic differentiation of different human hematopoietic precursors *in vitro* has been well established.⁵⁻⁷ Thus, it is important to identify the dose ranges of Am80 when combined with GCSF in mediating neutrophil differentiation to develop innate immunity against infection. Moreover, systematic review and meta-analysis of 5,256 patients show that giving GCSF to AML patients post-chemotherapy does not affect overall survival or infectious rate,⁸ whereas GCSF may induce myeloid malignancy in neutropenic patients.⁹⁻¹² Therefore, by testing different doses of Am80 when combined with GCSF for their effect on generating functional neutrophils while suppressing malignant growth in AML specimens, the defined dose ranges could have potential to serve as a reference baseline for an add-on therapy to GCSF in the future clinical study of Am80-GCSF for CCIN treatment. We have now titrated the combination of 25 ng/ml GCSF with different Am80 doses ranged from low to high human plasma

concentrations converted from clinical usages, including 20, 50, 100, and 150 nM Am80. These different combinations were evaluated for their *in vitro* effects on generating functional neutrophils while suppressing leukemic growth in primary AML patient specimens. The resultant data have now been provided in a new Fig. 3, whereas some other changes have been made in original Fig. 2, correspondingly. All these revisions are now presented in the revised manuscript (Results: page 7, 2nd paragraph, lines 3 to page 9, lines 1 to 10).

References

- ¹ Skubitz, K. M., Campbell, K. D. and Skubitz, A. P., CD66a, CD66b, CD66c, and CD66d each independently stimulate neutrophils, *J Leukoc Biol* 60, 106 (1996).
- ² Skubitz, K. M. and Skubitz, A. P., Interdependency of CEACAM-1, -3, -6, and -8 induced human neutrophil adhesion to endothelial cells, *J Transl Med* 6, 78 (2008).
- ³ Ding, W. et al., Retinoid agonist Am80-enhanced neutrophil bactericidal activity arising from granulopoiesis *in vitro* and in a neutropenic mouse model, *Blood* 121, 996 (2013).
- ⁴ Gresham, H. D. et al., Survival of *Staphylococcus aureus* inside neutrophils contributes to infection, *J Immunol* 164, 3713 (2000).
- ⁵ Hao, Q. L., Smogorzewska, E. M., Barsky, L. W. and Crooks, G. M., *In vitro* identification of single CD34+CD38- cells with both lymphoid and myeloid potential, *Blood* 91, 4145 (1998).
- ⁶ Luo, P. et al., Intrinsic retinoic acid receptor alpha-cyclin-dependent kinase-activating kinase signaling involves coordination of the restricted proliferation and granulocytic differentiation of human hematopoietic stem cells, *Stem Cells* 25, 2628 (2007).
- ⁷ Lou, S. et al., The lost intrinsic fragmentation of MAT1 protein during granulopoiesis promotes the growth and metastasis of leukemic myeloblasts, *Stem Cells* 31, 1942 (2013).
- ⁸ Gurion, R. et al., Colony-stimulating factors for prevention and treatment of infectious complications in patients with acute myelogenous leukemia, *Cochrane Database Syst Rev* 6, CD008238 (2012).
- ⁹ Smith, R. E., Bryant, J., DeCillis, A. and Anderson, S., Acute myeloid leukemia and myelodysplastic syndrome after doxorubicin-cyclophosphamide adjuvant therapy for operable breast cancer: the National Surgical Adjuvant Breast and Bowel Project Experience, *J Clin Oncol* 21, 1195 (2003).
- ¹⁰ Rosenberg, P. S. et al., The incidence of leukemia and mortality from sepsis in patients with severe congenital neutropenia receiving long-term G-CSF therapy, *Blood* 107, 4628 (2006).
- ¹¹ Hershman, D. et al., Acute myeloid leukemia or myelodysplastic syndrome following use of granulocyte colony-stimulating factors during breast cancer adjuvant chemotherapy, *J Natl Cancer Inst* 99, 196 (2007).
- ¹² Beekman, R. and Touw, I. P., G-CSF and its receptor in myeloid malignancy, *Blood* 115, 5131 (2010).

2nd Editorial Decision

01 September 2016

Thank you for the submission of your revised manuscript to EMBO Molecular Medicine. We have now received the enclosed reports from the referees that were asked to re-assess it. As you will see the reviewers are now globally supportive and I am pleased to inform you that we will be able to accept your manuscript pending the following final amendments:

- 1) Please address referee 1 concerns: we strongly recommend that you provide new light microscopy figures as suggested, rewrite according to this referees recommendations and have a native english speaker thoroughly go through the text to improve the english quality.

Please submit your revised manuscript within two weeks. I look forward to seeing a revised form of your article.

***** Reviewer's comments *****

Referee #1 (Comments on Novelty/Model System):

1) I am still not convinced by the image quality. The authors have simply run their original images through some kind of image restoration/contrast enhancement. The original images are the same and other than stated by the authors, running an image restoration algorithm on the same data does not improve their resolution. It is required to obtain images of typical H&E stains, where a nucleus is dark blue/violet against a light blue cytoplasm. Taking color photos through a good 63x NA 1.4 oil lens would provide perfectly resolved, hematology textbook quality images. This criticism applies to all provided light microscopy images.

Referee #1 (Remarks):

Major:

1) I am still not convinced by the image quality. The authors have simply run their original images through some kind of image restoration/contrast enhancement. The original images are the same and other than stated by the authors, running an image restoration algorithm on the same data does not improve their resolution. It is required to obtain images of typical H&E stains, where a nucleus is dark blue/violet against a light blue cytoplasm. Taking color photos through a good 63x NA 1.4 oil lens would provide perfectly resolved, hematology textbook quality images. This criticism applies to all provided light microscopy images.

2) Authors state in their revision that "the numbers of killed bacteria by Am80-GCSF mice were relatively small". I am not sure they understand their own data correctly. The Y-axis of this graph shows "Bacteria (10^4 /ml)" and the graph shows $\sim 0.8 \cdot 10^4$ /ml for A+G and $\sim 1.4 \cdot 10^4$ /ml for G-CSF. Control is $5.9 \cdot 10^4$ /ml. So, within 3 h neutrophils have killed 88 or 76% of all bacteria. This is not really a little, this is almost all of the bacteria. What is small, is the difference between A+G and G-CSF. This is what I referred to.

3) The English is still very poor. Also in the (helpful) newly added text. It should be strongly improved.

Referee #3 (Comments on Novelty/Model System):

My queries from first review have been answered, and so I think it is publishable

2nd Revision - authors' response

18 September 2016

Editor's recommendations and our responses

1. "have a native english speaker thoroughly go through the text to improve the english quality."

Response

We apologize for any existing language problems. The native English speakers, both Dr. David Warburton (a co-author of this manuscript and influential scientist/leader in Regenerative Medicine) and Dr. Martin Broome with expertise in Medical Biology, have now thoroughly reviewed the text, and edited the manuscript to improve the English quality.

Reviewer 1's comments and our responses

1. "I am still not convinced by the image quality. The authors have simply run their original images through some kind of image restoration/contrast enhancement. The original images are the same and other than stated by the authors, running an image restoration algorithm on the same data does not improve their resolution. It is required to obtain images of typical H&E stains, where a nucleus is dark blue/violet against a light blue cytoplasm. Taking color photos through a good 63x NA 1.4 oil lens would provide perfectly resolved, hematology textbook quality images. This criticism applies to all provided light microscopy images."

Response

We agree with the reviewer that the quality of the images could be better. However, we respectfully disagree with the reviewer's comments on the staining method. We believe that Giemsa stain is a better stain than H&E stain for the purpose of this study.

Giemsa stain, also called differential stain, is specific for the phosphate groups of DNA and thus distinctively blots the regions of DNA. It is a classic blood film stain for peripheral blood (PB) smears and bone marrow (BM) specimens. To date, Giemsa stain has been widely applied as a standard method for evaluating neutrophil morphologic differentiation through assessing the degrees of neutrophil nuclear segmentation (Gallagher et al, 1979; Ding et al, 2013). This was why, as with many other groups, we used Giemsa stain of PB and BM cells for determining changes in neutrophil nuclear segmentation.

We have now provided new light microscopy images derived from Giemsa stain. We did our best to capture new images with better quality to clearly distinguish nucleus and cytoplasm. We believe that these new images allow for evaluating the degrees of neutrophil morphologic differentiation reflected by neutrophil nuclear segmentation in more mature vs. less mature neutrophils.

2. "Authors state in their revision that "the numbers of killed bacteria by Am80-GCSF mice were relatively small". I am not sure they understand their own data correctly. The Y-axis of this graph shows "Bacteria ($10^4/ml$)" and the graph shows $\sim 0.8 \cdot 10^4/ml$ for A+G and $\sim 1.4 \cdot 10^4/ml$ for G-CSF. Control is $5.9 \cdot 10^4/ml$. So, within 3 h neutrophils have killed 88 or 76% of all bacteria. This is not really a little, this is almost all of the bacteria. What is small, is the difference between A+G and G-CSF. This is what I referred to."

Response

We agree with the reviewer. Our data (Fig. 5C) showed that, compared to GCSF mice, bacterial killing in Am80-GCSF mice significantly increased more after 16 hr of infection ($P < 0.001$) than bacterial killing after 3 hr of infection ($P < 0.01$).

3. "The English is still very poor. Also in the (helpful) newly added text. It should be strongly improved."

Response

Please see our response to Editor's recommendations #1.

References

Gallagher R, Collins S, Trujillo J, McCredie K, Ahearn M, Tsai S, Metzgar R, Aulakh G, Ting R, Ruscetti F, and Gallo R (1979) Characterization of the Continuous, Differentiating Myeloid Cell Line (HL-60) From a Patient With Acute Promyelocytic Leukemia. *Blood*, **54**(3): 713-733

Ding W, Shimada H, Li L, Mittal R, Zhang X, Shudo K, He Q, Prasadarao NV, Wu L (2013) Retinoid agonist Am80-enhanced neutrophil bactericidal activity arising from granulopoiesis in vitro and in a neutropenic mouse model. *Blood* **121**(6): 996-1007

Corresponding Author Name: Lingtao Wu

Manuscript Number: EMM-2016-06434